# ☕ ESE: Espresso Sentence Embeddings

**Xianming Li** [1], **Zongxi Li** [2], **Jing Li** [1,3]*, **Haoran Xie** [2] , **Qing Li** [1,3]

[1] Department of Computing,
[3] Research Centre on Data Science & Artificial Intelligence,
**The Hong Kong Polytechnic University, Hong Kong SAR**

[2] School of Data Science,
**Lingnan University, Hong Kong SAR**

```
xianming.li@connect.polyu.hk,
{zongxili,hrxie}@ln.edu.hk,
{jing-amelia.li,qing-prof.li}@polyu.edu.hk
```

## ABSTRACT

High-quality sentence embeddings are fundamental in many natural language processing (NLP) tasks, such as semantic textual similarity (STS) and retrieval-augmented generation (RAG). However, most existing methods leverage fixed-length sentence embeddings from full-layer language models, which lack the scalability to accommodate the diverse available resources across various applications. Viewing this gap, we propose a novel sentence embedding model Espresso Sentence Embeddings (ESE) with two learning processes. First, the **learn-to-express** process encodes more salient representations to shallow layers. Second, the **learn-to-compress** process compacts essential features into the initial dimensions using Principal Component Analysis (PCA). This way, ESE can scale model depth via the former process and embedding size via the latter. Extensive experiments on STS and RAG suggest that ESE can effectively produce high-quality sentence embeddings with less model depth and embedding size, enhancing embedding inference efficiency. The code is available at `https://github.com/SeanLee97/AnglE/blob/main/README_ESE.md`.

## 1 INTRODUCTION

Sentence embedding learning (Cer et al., 2018; Reimers & Gurevych, 2019; Gao et al., 2021; Li & Li, 2024a;b) is a crucial yet challenging task in the NLP research. It aims to capture essential semantic and syntactic information in language, benefiting various scenarios such as clustering (Reimers & Gurevych, 2019), semantic textual similarity (STS) (Li & Li, 2023; Zhang et al., 2024), and retrieval-augmented generation (RAG) (Gao et al., 2023).

In the common deployment practices, the pipeline of applying sentence embeddings unfolds in two typical stages: (i) computing the sentence embeddings via a forward pass and (ii) employing these embeddings in downstream tasks. Existing work (Reimers & Gurevych, 2019; Gao et al., 2021; Li & Li, 2024a, *inter alia.*) typically adopts entire Transformer (Vaswani et al., 2017) layers and full embedding sizes for all tasks to ensure optimal performance, regardless of the varying resources and requirements across applications. It can result in computational redundancy and fails to scale well to the diverse resources available in downstream scenarios (Kusupati et al., 2022). To address this challenge, Matryoshka Representation Learning (MRL) concurrently trains multiple embeddings with cascading dimensions to enable scalable embedding sizes while preserving maximum semantics (Kusupati et al., 2022). However, MRL employs full Transformer layers for embedding inference, in which high computational costs persist when using Large Language Models (LLMs) with very deep architectures (Wang et al., 2023; Li & Li, 2024a;b; Lee et al., 2024).

---

* Corresponding Author.

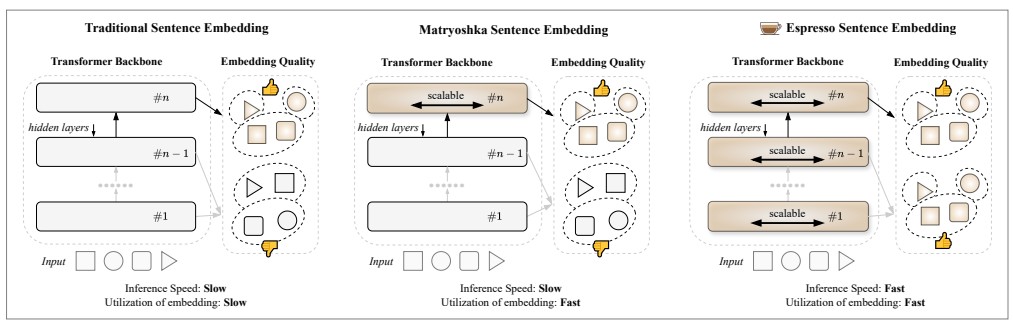

Figure 1: The comparison of traditional (left), MRL (middle) (Kusupati et al., 2022), and the proposed ESE (right) sentence embedding models. The gray blocks represent Transformer layers that are fine-tuned in the full model setting, while the coffee-colored blocks indicate layers used in scalable settings. In traditional and MRL approaches, while the final layer produces high-quality embeddings, the intermediate layers yield poor-quality embeddings. In contrast, the proposed ESE enables flexible scaling along both model depth and embedding dimension.

We propose a novel Espresso Sentence Embeddings (ESE) to further improve inference efficiency by enabling scalability on model depth. ESE consists of two key processes. First, the **learn-to-express** process allocates more crucial latent representations to shallow layers by weighting embeddings at different levels. Second, the **learn-to-compress** process condenses essential features into initial dimensions by exploring the inner dependencies of embedding dimensions through Principal Component Analysis (PCA). This mechanism allows the first-$k$-dimension sub-embeddings to exhibit robust performance, offering flexibility in dynamic settings with varying embedding capacity requirements but with guaranteed performance by adjusting the value of $k$. Figure 1 illustrates ESE's overview and its differences from the MRL (Kusupati et al., 2022) and traditional embeddings (Reimers & Gurevych, 2019; Gao et al., 2021; Li & Li, 2024a). As depicted, ESE is designed to encode more salient features into shallow layers and small embedding dimensions simultaneously. Thus, it enables scalable sentence embeddings that adapt to various settings of model depth and embedding size, allowing for more flexible accommodation of diverse computing resources. In contrast, MRL focuses solely on scaling embedding size, while traditional models lack scalability. Additionally, the PCA implementation of ESE at various layers of embedding learning can help organize the learned features in order according to their significance; it allows easier training than MRL by jointly training multiple varying-dimension embeddings to scale embedding sizes.

To the best of our knowledge, *we are the first to learn sentence embedding with information compression, presenting scalable embedding inference to both model depths and embedding dimensions.*

We extensively experiment across STS and RAG tasks to evaluate ESE. First, the main results on the STS benchmarks show that ESE performs competitively in full settings than non-trivial baselines and shows significantly better results with shallow model depth or smaller embedding size. For instance, ESE enhances BGE's (Xiao et al., 2023) shallow-layer embeddings from 45.60 to 66.27, showcasing the effectiveness of our method. Then, ablation studies indicate that all modules positively affect ESE, and its performance is sensitive to smaller PCA compression sizes. Moreover, the RAG experiments show that ESE improves retrieval across varying embedding sizes and model depths, underscoring the potential in various application scenarios. Finally, we further discuss ESE and present the following findings: (i) ESE shows better inference efficiency on both STS and RAG tasks; (ii) In visualization, ESE's scaled embeddings exhibit greater overlap with the unscaled ones, indicating its effectiveness in information compression and superiority in scaling embeddings.

In summary, our contributions are as follows:

• We are the first to employ the information compression technique to scale sentence embeddings.

• Our novel ESE model allows for scalable embeddings in both model depth and embedding size.

• Extensive experiments on STS and RAG show ESE's superiority in producing effective embeddings with reduced model depth and embedding size, enhancing inference efficiency.

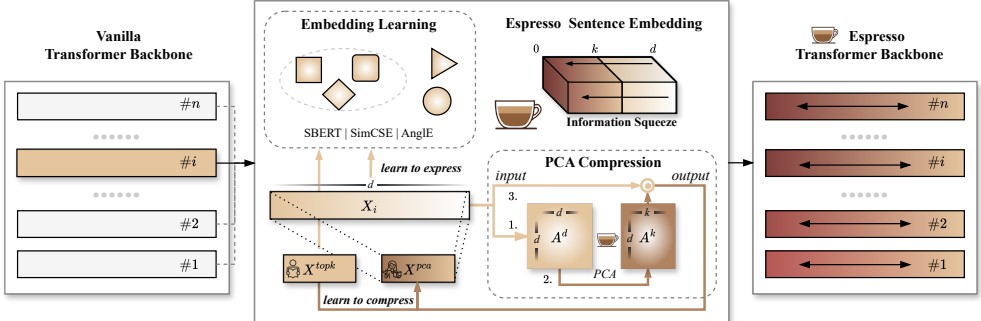

Figure 2: The ☕ ESE framework. The left part is the vanilla Transformer backbone, where each layer is not scalable. The center part is the ESE training with the **learn-to-express** (to scale model depths) and **learn-to-compress** (to scale embedding sizes) processes. The right part is the trained Espresso Transformer backbone, where each layer is scalable (marked with double-ended arrows).

## 2   RELATED WORK

Our work is in line with the sentence embedding learning research. While early efforts primarily focused on word embeddings (Mikolov et al., 2013), the recent trend has shifted towards sentence embeddings as they can capture representations from richer contexts. In the training methods, many studies adopted supervised approaches to align embeddings to human senses (Conneau et al., 2017; Cer et al., 2018; Reimers & Gurevych, 2019; Li & Li, 2024a). And **contrastive learning** techniques (Carlsson et al., 2020; Gao et al., 2021; Chuang et al., 2022; Xu et al., 2023; Liu, 2024) have recently become popular to involve in-batch comparison of positive and negative pairs. As for model architectures, with the LLMs breakthrough in NLP (OpenAI, 2022; Touvron et al., 2023), **LLM-based** sentence embedding models have drawn a growing attention (Li & Li, 2024a; Wang et al., 2023; Li & Li, 2024b; Muennighoff et al., 2024) for a more effective context exploration.

However, the common practices deploy embeddings with full model layers and predefined embedding sizes. It inevitably constrains the scalability of sentence embedding use in downstream applications, particularly in scenarios where resources are limited. To address this constraint, Matryoshka Representation Learning (MRL) was introduced to allow scalable embedding sizes (Kusupati et al., 2022). Yet, it still relies on full Transformer layers for embedding inference. In contrast, ESE has dual scalability in both model depth and embedding size, largely enhancing the efficiency of embedding inference. Moreover, MRL trains multiple embeddings in varying sizes, whereas ESE employs PCA for embedding size compression. Here, the PCA in training can help organize the features in order and allow more effective embedding learning. While some other methods (Zhu et al., 2018; Gupta et al., 2019; Zhao et al., 2022) applied PCA-alike compression to pre- or post-process trained embeddings, ESE is the first to integrate PCA into sentence embedding training.

## 3   ☕ ESPRESSO SENTENCE EMBEDDINGS

This section describes how ESE learns scalable sentence embeddings by elaborating the learn-to-express and learn-to-compress processes. Figure 2 depicts its framework.

### 3.1   ENCODER

We start the discussion with how we process the input, where the pretrained language model are used as an encoder to transform the text into dense sentence embeddings. Here, BERT (Devlin et al., 2019) or LLMs (Touvron et al., 2023; Bai et al., 2023) serves as the backbone to encode text $x$ as follows:

$$\mathbf{X}_i^k = \text{Pooling}(\text{Encoder}^{1:i}(x)) \in \mathbb{R}^{k \times 1}, \tag{1}$$

where $i \in [1, n]$ is first-$i$ layers of the $n$-layer Transformer backbone, and $k \in [1, d]$ denotes first-$k$-dimensional sub-embeddings of the full $d$-dimensional embeddings. For BERT-based models,

we use the "CLS" pooling strategy following Li & Li (2024a). For LLM-based models, we follow previous practices (Li et al., 2023; Li & Li, 2024b) to transform all layers into bidirectional and adopt the mean pooling. Here, we select BERT and LLMs like Qwen and LLaMA as the encoder since they are widely used Transformer-based models for sentence embeddings. It allows for evaluating the effectiveness and generalization of the proposed ESE across various model scales.

## 3.2 Learn to Express

The encoded embeddings $\mathbf{X}_i^k$ will then go through a **learn-to-express** process to allow scalable model depths. Our idea is to encode more essential representations into shallow layers. Concretely, we cache each layer's sentence embeddings obtained by Eq. 1 for $i \in [1, n-1]$ and then jointly train their first-$k$-dimension sub-embeddings by a weighted loss. The loss is computed as follows:

$$\mathcal{L}_{le} = \sum_{i=1}^{n-1} w_i * \text{loss}(\mathbf{X}_i^k, \mathcal{G}) + \text{loss}(\mathbf{X}_n^k, \mathcal{G}). \tag{2}$$

Here $w_i$ denotes the weight for $i$-th layer. We provide two implementations for the weights $w_i$. The first implementation is to parameterize $w_i$ so that the weights can be adjusted dynamically during the model training. The second implementation sets $w_i = \frac{1}{1+\ln(i)}$. This way, the weights decrease as the layer depth increases, allowing the shallow layers to capture more crucial information for embedding scalability in model depths. Note that we leave the last layer unweighted due to its critical role in capturing sentence-level semantics (Li & Li, 2024b). The second implementation is applied by default as it does not introduce any additional parameters, making the implementation more efficient as the efficiency is always an important factor for the proposed ESE. The $\text{loss}(\cdot)$ can be any loss function for sentence embedding learning, e.g., contrastive loss (Gao et al., 2021) or AnglE loss (Li & Li, 2024a). We use the latter one by default. $\mathcal{G}$ is the positive or negative sample indicators for loss computation.

## 3.3 Learn to Compress

The learn-to-express method (Section 3.2) focuses on scaling embeddings across layers for model depths. Then, the **learn-to-compress** aims to enable scalable embedding sizes within each layer while preserving maximum semantics. The goal is to allow ESE to condense more essential information into sub-vectors of the initial embedding dimensions, as shown in the "Information Squeeze" box in Figure 2. It shows information distributions, where the left-hand-side part contains higher "information density" and a richer representation than the right-hand-side.

To achieve the above, we use Principal Component Analysis (PCA) to reduce embeddings' dimensionality from size $d$ to $k$ while preserving the crucial semantics, where $d$ and $k$ denote full and compressed embedding sizes, respectively. We compress the inner dependencies within embedding dimensions instead of directly compressing the sentence embeddings. This step aims to enable ESE to capture nuanced cross-dimension interactions without oversimplifying the embedding structure.

Concretely, we first compute the embedding inner-dependency matrix by scaled dot product:

$$\mathbf{A}_i^d = \text{softmax}(\frac{\mathbf{X}_i^d \cdot \mathbf{X}_i^{d^T}}{\sqrt{d}}) \in \mathbb{R}^{d \times d}, \tag{3}$$

where $i$ denotes first-$i$ Transformer layers. $\mathbf{X}_i^d \in \mathbb{R}^{d \times 1}$ are full sentence embeddings obtained from Eq. 1. The design pertains to the self-attention of Transformer Vaswani et al. (2017) reflecting the weights across varying dimensions. Then, we employ PCA, implemented with Singular Value Decomposition (SVD), to compress its information. It thus results in $\mathbf{A}_i^d = \mathbf{U}_i \mathbf{\Sigma}_i \mathbf{V}_i^T$, based on which we obtain top-$k$ principal components (i.e. dependencies) for each embedding, as follows:

$$\mathbf{A}_i^k = \mathbf{U}_i^{1:d,1:k} \mathbf{\Sigma}_i^{1:k,1:k} \in \mathbb{R}^{d \times k}, \tag{4}$$

where $1 \leq k \leq d$. $1:d, 1:k$ means the selection of the first-$d$ rows and first-$k$ columns, and $\mathbf{A}_i^k$ indicates the compressed inner-dependency matrix. The diagonal matrix $\mathbf{\Sigma}_i$ displays the singular values $\sigma_{i,1}, \sigma_{i,2}, \ldots, \sigma_{i,k}$ on its diagonal, arranged in descending order according to their magnitude.

Subsequently, we apply the top-$k$ inner-dependency matrix $\mathbf{A}_i^k$ (with compressed self-attention-alike weights) to the original $\mathbf{X}_i^d$. It results in compressed sentence embeddings as follows:

$$\mathbf{X}_{i}^{k} = \mathbf{A}_i^{k^T} \cdot \mathbf{X}_i^d \in \mathbb{R}^{k \times 1}. \tag{5}$$
$$\underset{pca}{}$$

Although the above process works only in training, PCA may cause higher computational costs in embedding inference. To tackle this issue, we align the truncated first $k$-dimensional sub-embeddings to the PCA-compressed ones to reduce the cost. The alignment involves mean squared error and Kullback-Leibler divergence losses (Kim et al., 2021).

As an innovative approach, we employ a weighted strategy across layers, in alignment with the learn-to-express process, to coherently coordinate the compression of embedding sizes across scalable model depths, as follows:

$$\mathcal{L}_{lc} = \sum_{i=1}^{n-1} w_i * \text{align}(\mathbf{X}_i^k, \underset{pca}{\mathbf{X}_i^k}) + \text{align}(\mathbf{X}_n^k, \underset{pca}{\mathbf{X}_n^k}), \tag{6}$$
$$\text{align}(x, y) = \text{MSE}(x, y) + \text{KLDiv}(x, y),$$

where $\text{MSE}(\cdot)$ and $\text{KLDiv}(\cdot)$ denote the mean squared error and Kullback-Leibler divergence, respectively. $w_i$ means the weight for the $i$-th layer. It has the same setting as Eq. 2. By optimizing $\mathcal{L}_{lc}$, the truncated first $k$-dimensional sub-embedding will be aligned to the PCA-compressed ones. It allows the direct use of the first $k$-dimensional sub-embeddings at inference without repeatedly performing PCA and improves inference efficiency.

### 3.4 Joint Learning

Finally, we jointly train the learn-to-express and learn-to-compress processes within a unified framework. Their respective objectives are seamlessly integrated into the overall learning objective of the proposed ESE, as follows:

$$\mathcal{L} = \alpha \mathcal{L}_{le} + \beta \mathcal{L}_{lc}, \tag{7}$$

where $\alpha$ and $\beta$ are the weights trading off the two processes. We set both to 1 by default. By jointly optimizing the two objectives, ESE trains its sentence embeddings to be scalable in both model depths and embedding sizes.

## 4 Experiment

In the experiments, we first present the main intrinsic results of STS in Section 4.1. Then, we probe into more details of ESE output via an ablation study in Section 4.2. Finally, we further discuss ESE from varying perspectives in Section 4.4 to provide more insight.

### 4.1 STS Experiments

Following common practices (Gao et al., 2021; Li & Li, 2024a), we adopt STS for intrinsic evaluation to assess the quality of trained sentence embeddings. It measures how well embeddings capture semantic similarity between sentences, indicating their capacity for effective text representation.

**Setup.** In the comparison, we include two baselines: the scalable embeddings MRL (with scalable embedding sizes) (Kusupati et al., 2022) and RAW embeddings (without any scaling operations).

For the *datasets*, we train sentence embeddings on MultiNLI (Williams et al., 2018) and SNLI (Bowman et al., 2015) datasets following previous studies. Evaluation of model performance is conducted on the STS benchmark computed by SentEval (Conneau & Kiela, 2018), where Spearman's correlation in the "all" setting is reported as the evaluation metric. To enable comprehensive evaluation, this benchmark comprises seven widely used STS datasets: STS 2012-2016 (Agirre et al., 2012; 2013; 2014; 2015; 2016), SICK-R (Marelli et al., 2014), and STS-B (Cer et al., 2017).

For *model settings*, we examine two popular BERT-based backbones, *bge-base-en-v1.5* (Xiao et al., 2023) and *UAE-Large-V1* (Li & Li, 2024a) , and an LLM-based backbone, *Qwen1.5* (Bai et al.,

Table 1: STS benchmark results. The last column ($\prec$ Avg.) is the average results of shallow layers (except the last one), while the remaining correspond to the last-layer results. Avg.: average results over varying benchmark datasets. RAW: the original model; MRL: Kusupati et al. (2022). The coffee-colored cells: the best results for each backbone model; boldfaced numbers: the overall best results. For $\prec$ Avg., ESE performs significantly better than baselines: $p$-value $< 0.05$ (paired t-test).

| Model | STS12 | STS13 | STS14 | STS15 | STS16 | STS-B | SICK-R | Avg. | $\prec$ Avg. |
|---|---|---|---|---|---|---|---|---|---|
| **bge-base-en-v1.5** (Xiao et al., 2023) | | | | | | | | | |
| RAW | 78.03 | 84.18 | 82.27 | 87.96 | 85.47 | 86.41 | 79.88 | 83.46 | 45.60 |
| + MRL | 75.90 | 87.87 | 83.97 | 88.92 | 85.07 | 87.17 | 79.18 | 84.01 | 46.18 |
| + ESE | 77.70 | 86.97 | 83.57 | 89.43 | 86.16 | 87.27 | 80.32 | 84.49 | **66.27** |
| **UAE-Large-V1** (Li & Li, 2024a) | | | | | | | | | |
| RAW | 79.09 | 89.62 | 85.02 | 89.51 | 86.61 | 89.06 | 82.10 | 85.86 | 44.80 |
| + MRL | 78.26 | 90.19 | 84.91 | 89.48 | 86.17 | 88.49 | 79.28 | 85.25 | 44.97 |
| + ESE | 79.64 | 90.40 | 85.76 | 90.33 | 86.64 | 88.54 | 81.09 | 86.06 | 59.12 |
| **Qwen1.5-0.5B** (Bai et al., 2023) | | | | | | | | | |
| RAW | 75.91 | 83.77 | 80.04 | 86.05 | 82.91 | 85.32 | 78.98 | 81.85 | 56.59 |
| + MRL | 76.30 | 85.04 | 80.68 | 86.15 | 83.12 | 85.65 | 79.45 | 82.34 | 58.22 |
| + ESE | 76.43 | 85.70 | 81.75 | 86.30 | 83.67 | 85.76 | 80.16 | 82.82 | 59.99 |

2023). We use the recent popular AnglE (Li & Li, 2024a) loss as the sentence embedding training loss. The initial learning rates are set to $5e-5$ and $2e-4$ to train BERT-based and LLM-based models, respectively. For efficient LLM fine-tuning, we utilize LoRA (Hu et al., 2021; Dettmers et al., 2024) with parameters $lora\_r = 32$, $lora\_alpha = 32$, and $lora\_dropout = 0.1$. For the ESE setup, the compression dimension $k$ is set to 128 by default, and the weights $\alpha$ and $\beta$ for joint learning (Eq. 7) are set to 1.

**Main Results** In the main STS experiments, we extract embeddings from the last 12 Transformer layers for comprehensive analyses. Then, we follow common practices to report results from the last layer of all benchmark datasets, including their average (Avg.). Moreover, we report the average results from the shallow layers (excluding the last one) to assess the scalability of ESE across model depths ($\prec$ Avg.). The results are shown in Table 1, where we can draw the following observations.

First, ESE shows competitive and even better performance compared to RAW across every backbone in the last layer results. It indicates that the scalability learning of ESE does not adversely impact its last-layer performance. The marginal performance gain could be attributed to the use of PCA in organizing features, which potentially facilitates more effective embedding learning even in an unscaled setup. In contrast, MRL fails to outperform RAW in UAE's last layer, possibly due to chaos introduced by its multi-embedding learning strategy. Second, ESE significantly outperforms baselines in the $\prec$ Avg. setup , with a 20.67 gain to BGE. It indicates ESE's capability of effectively scaling embeddings across different layers. The improvement is less notable in LLM-based models than in BERT-based models, possibly due to LoRA's limitation on updating parameters in LLMs.

To further examine the scalability of ESE in model depths and embedding sizes, we show the STS results over embedding sizes from every layer in Figure 3. While all the models achieve competitive results using the full-sized embeddings from the last layer, RAW and MRL exhibit inferior performance in the shallow layers. They even demonstrate performance fluctuations as the layers deepen. In contrast, ESE can significantly improve the performance of the shallow layers. For example, the BGE achieves a result over 60.00 at the 11th layer, while BGE-MRL requires ten layers to reach that level. In contrast, BGE-ESE only requires two layers and consistently improves until the last layer, reaching a score of 84.49. This shows the effectiveness of ESE's learn-to-express process in encoding the more important information into shallow layers.

Moreover, ESE consistently performs better over embedding sizes at each layer, indicating the effectiveness of its learn-to-compress process. The performance gain is more significant at shallow layers, potentially because they contain richer information, resulting in more effective compression. Notably, ESE's stable performance since the 128 dimension suggests that a 128-dimensional embedding may capture the majority of salient features.

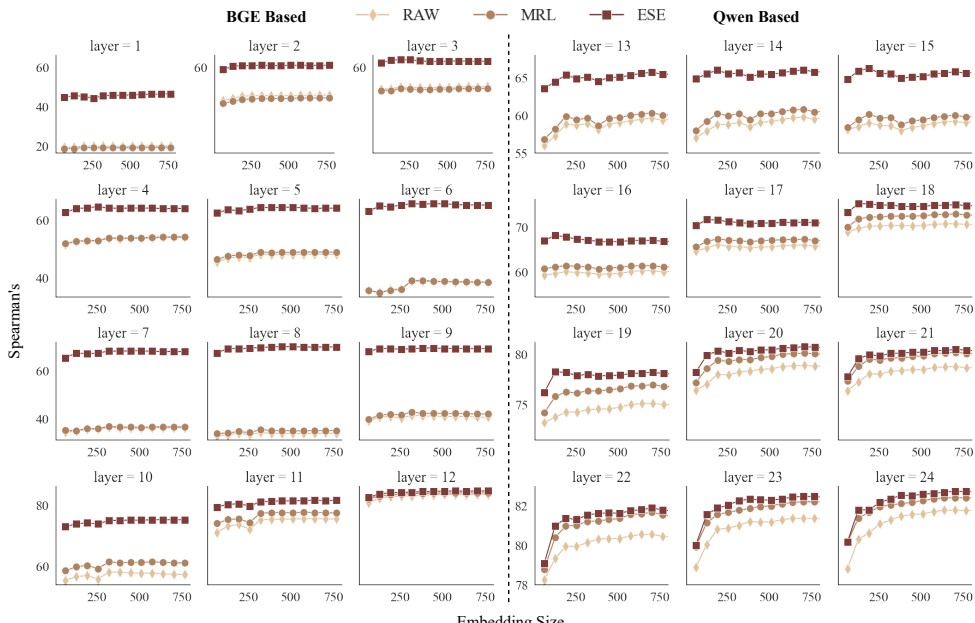

Figure 3: Results of the STS benchmark for each of the last 12 layers of BGE-based backbone (left part) and Qwen-based backbone (right part). For each layer's result, the x-axis shows the embedding size, and the y-axis shows the average Spearman's correlation over varying benchmark datasets.

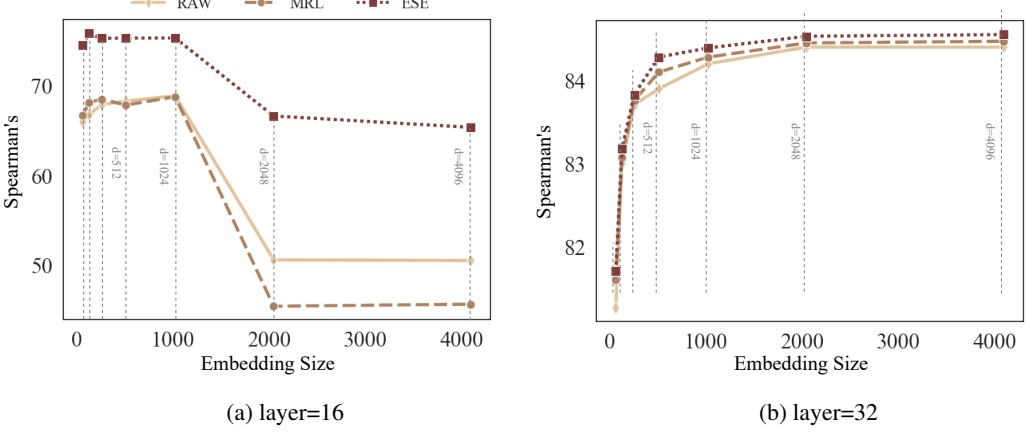

(a) layer=16

(b) layer=32

Figure 4: The average STS benchmark results of various embedding sizes for the half layer (first 16 layers) and last layer (full 32 layers) of LLaMA2 7B (Touvron et al., 2023) with the RAW, MRL, and ESE training.

Furthermore, considering the *Qwen1.5* 0.5B is a relatively small LLM (Lu et al., 2024), we also evaluate the effectiveness of ESE on larger LLMs by testing LLaMA2 7B (Touvron et al., 2023) on the STS benchmark, as shown in Figure 4. Here, we report the results of two typical layers, i.e. the middle layer (layer=16) and the last layer (layer=32). It is evident that the proposed ESE consistently outperforms baselines across different model depths and embedding sizes. Notably, ESE achieves a significant improvement at shallow layers, demonstrating its superior scalability. We can also see that there is a performance drop observed at the middle layer for baselines and ESE. This could be attributed to the LoRA's limitation in sentence embedding learning. Nevertheless, ESE can mitigate this limitation, as evidenced by its lower drop rate than baselines.

## 4.2 Ablation Study

While Section 4.1 shows the overall effectiveness of ESE, we conduct ablation studies of ESE on the STS benchmark to assess the effectiveness of each component. The results are presented in Table 2.

We first evaluate different compression strategies to scale embedding sizes: the default PCA on embedding dependencies, PCA on sentence embeddings directly (i.e., PCA on $\mathbf{X}_i^d$), the max-$k$ embedding dependencies (i.e., max $k$ of dependency matrix $\mathbf{A}_i^d$ in Eq. 3 without PCA), and the ablation without any compression. All components contribute positively to the learning of both the last-layer and the shallow-layer embeddings. Interestingly, the performance drops substantially without compression, suggesting that compression can also help the learning of scaling across layers.

As to the weighting strategies, we compare the performance of without, dynamic, and default layer weighting. The results indicate that without layer weight performs worse than dynamic and default in both the last layer and shallow layer learning, particularly the latter. It suggests that assigning different layer weights allows shallow layers to capture more essential information. While the dynamic layer weighting strategy outperforms the default strategy in the last layer, it performs worse for shallow layers. This might be because dynamic weighting places more importance on the last layer of learning, which negatively affects the shallow layers' performance, thereby hurting the scalability of model depth. The default layer weighting strategy consistently improves the performance of both the last and shallow layers. This could be because the performance improvements in the shallow layers propagate to the deeper layers, resulting in an overall performance improvement.

Furthermore, we assess the impact of different PCA compression sizes ($k$) during training. We train ESE on NLI datasets with varying compression sizes and evaluate their STS performance in the full setting. For comparison with the baselines, we report ESE's and baseline's results in Figure 5 [1]. The results indicate that ESE consistently outperforms the baselines across different compression sizes, highlighting the effectiveness of information compression in enhancing sentence embeddings. However, there exhibits a first-increase-then-decrease trend (peaked at 128). This is because the model with large compression sizes may overlook critical details, whereas that with small compression sizes might not have adequate capacity to retain all essential information.

Table 2: Ablation study of ESE on STS benchmark. Avg. and $\prec$ Avg. denote the average Spearman's correlation of the last layer and shallow layers (except the last), respectively.

| Model | Avg. | $\prec$ Avg. |
|---|---|---|
| *Compression Strategies* | | |
| PCA on dependencies (default) | **84.49** | **66.27** |
| PCA on sentence embeddings | 84.30 | 65.69 |
| max-$k$ embedding dependencies | 84.22 | 65.47 |
| none (w/o compression) | 84.13 | 50.18 |
| *Layer Weighting Strategies* | | |
| without weight | 84.25 | 64.56 |
| dynamic weight | **84.63** | 65.94 |
| default weight | 84.49 | **66.27** |

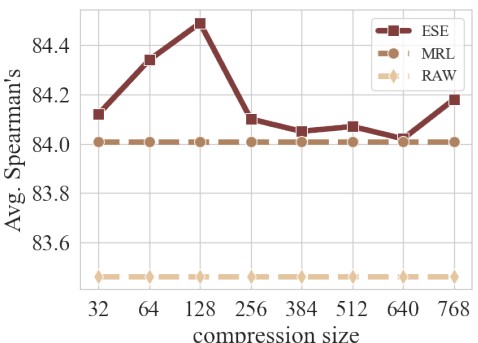

Figure 5: The STS benchmark results of ESE trained on different PCA compression sizes. Dashed lines indicate baseline results, serving as reference points.

## 4.3 RAG Experiments

The previous discussions center on the intrinsic evaluation in STS. We conduct an extrinsic evaluation to examine how ESE helps downstream retrieval and generation deployment. We compare different sentence embedding models based on *bge-base-en-v1.5* (Xiao et al., 2023) on HotpotQA dataset. It contains 113k Wikipedia factoid question-answer pairs. We employ faiss (Johnson et al.,

---

[1]The Raw setting does not support embedding compression, and MRL supports embedding truncation rather than compression. Hence, we depict their highest performance as light-colored lines in Figure 5.

2019) to index and retrieve documents based on sentence embeddings. Since our sentence embeddings are normalized, cosine similarity or Euclidean distance can be used. We follow the common practice (Douze et al., 2024) of using Euclidean distance, i.e., IndexFlatL2 in faiss, as the similarity measurement. For each query, we retrieve the most relevant context from all Wikipedia documents in HotpotQA and generate responses using LLaMA2 7B (Touvron et al., 2023) via a prompt in Table 4. For the QA evaluation, we adopt the popular metric *Exact Match*.

The RAG results of the last layer with various embedding sizes are listed in Table 3. We observe that ESE outperforms baselines in all embedding sizes. The improvements are more evident with smaller embedding sizes, for example, $2.42\%$ and $1.80\%$ higher than RAW and MRL with $64$-dimensional embedding, respectively. Moreover, we compare the performance when using the sentence embeddings from the half layer ($n = 6$). ESE shows $0.72\%$ and $0.44\%$ improvement compared to RAW and MRL, respectively. These results demonstrate that effective ESE sentence embeddings, scalable in terms of both embedding sizes and model depths, can further benefit RAG.

Table 3: RAG exact matching scores with different embedding sizes of different sentence embedding models on HotpotQA. *bge-base-en-v1.5* serves as the backbone sentence embedding model (marked with RAW).

| embedding size | Model | | |
|---|---|---|---|
| | RAW | + MRL | + ESE |
| 64 | 29.86 | 30.48 | **32.28** |
| 128 | 38.85 | 38.90 | **39.28** |
| 256 | 42.05 | 42.20 | **42.50** |
| 512 | 44.16 | 44.20 | **44.44** |
| 768 | 45.06 | 45.09 | **45.31** |

Table 4: The prompt used in LLaMA2 7B for the RAG task. The "{context}", and "{question}" are the placeholders for the input context and question.

As an advanced QA system, your role is to provide accurate and straightforward answers based on the provided context. Utilize the following information to answer the given question, directly output the straightforward answer, and do not explain it:

**Context:** {context}.

**Question:** {question}.

**Requirement:** For 'yes/no' question, directly output 'yes' or 'no'. For the 'who' question, directly output the people's names. For the 'when' question, directly output the date or time, so on and so forth.

**Output:**

### 4.4 DISCUSSION

**Efficiency.** We compare ESE with MRL (Kusupati et al., 2022) in terms of efficiency and effectiveness on STS, as shown in Figure 6a. Although the inference times at different layers are largely the same, the performance gaps between MRL's and ESE's embeddings are significant. This shows that our approach can effectively improve the shallow layer's task-specific representation, underscoring the scalability of the ESE model. We also examine how ESE affects RAG efficiency. Figure 6b illustrates the time consumed by the encoding and retrieval stages of the RAG pipeline with various embedding sizes. The results indicate that smaller embedding sizes are more efficient than larger ones, and ESE is slightly more efficient than baselines. It is because the ESE's high-quality embeddings in varying scaled sizes allow easier and more effective indexing, improving RAG efficiency.

**Compression Quality.** To intuitively compare the information compression quality between ESE and baselines, we use t-SNE (Van der Maaten & Hinton, 2008) to visualize 768-dimensional (full) and 128-dimensional (scaled) sentence embeddings in 2D space, as shown in Figure 7. ESE exhibits a higher overlapping rate between the dark- and light-color dots than baselines, showing ESE's high fidelity in the compressing process, i.e., ESE's first-128 sub-embeddings carry most key information from their full embeddings. To quantitatively measure the difference, we calculate the average

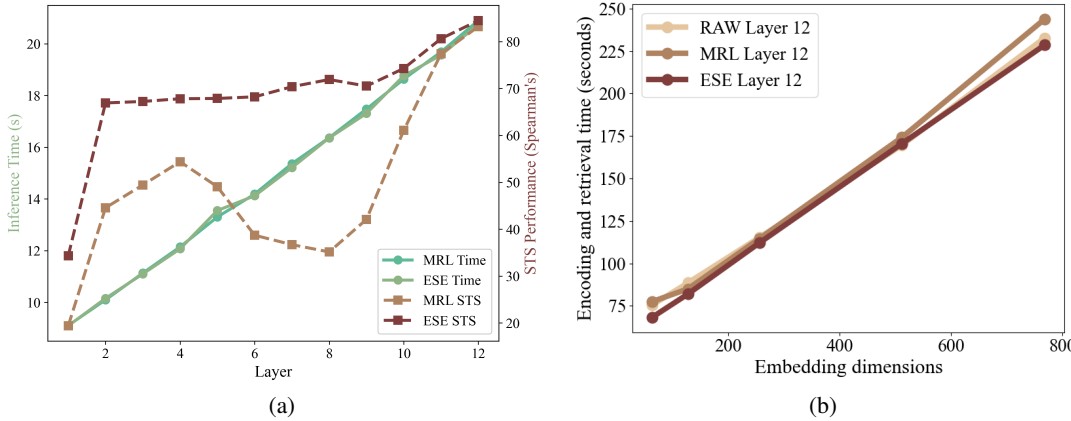

(a)  (b)

Figure 6: (a) Inference time and STS performance *vs* layers. (b) The encoding and retrieval time with different embedding sizes on the HotpotQA dataset.

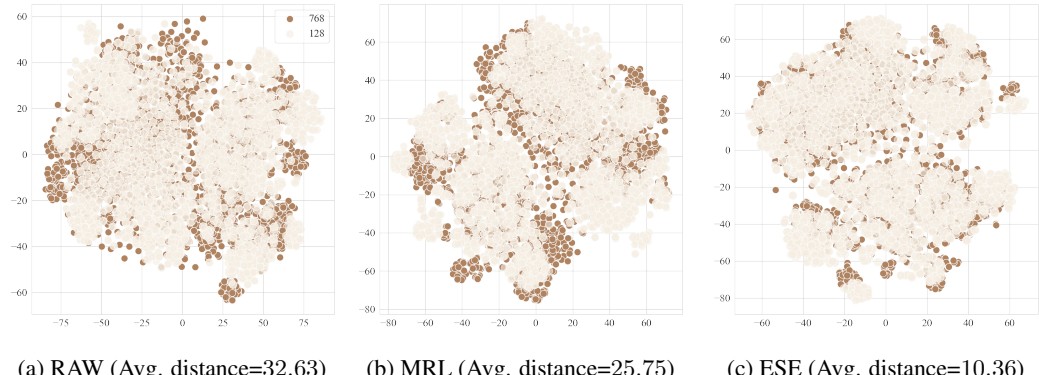

(a) RAW (Avg. distance=32.63)  (b) MRL (Avg. distance=25.75)  (c) ESE (Avg. distance=10.36)

Figure 7: The t-SNE visualization of sentence embeddings on the STS-B test set. *bge-base-en-v1.5* serves as the backbone for RAW, MRL, and ESE models. The dark-color dots denote the full 768-dimensional sentence embeddings, while light-color ones represent the first 128-dimension sub-embeddings. The Avg. means the average distance between the 768-dimensional and the 128-dimensional embedding points.

distance between the 768- and 128-dimensional embedding points and put the numbers in Figure 7. It is evident that the proposed ESE achieves the lowest average distance among the baselines. This indicates that ESE can effectively condense essential information into the initial embedding dimensions by the learn-to-compress process. It can reduce information loss when using small first-$k$ sub-embeddings for downstream applications.

## 5 CONCLUSION

In this paper, we have presented a novel sentence embedding model called ☕ Espresso Sentence Embeddings (ESE). It is the first work to support both model depth and embedding size scaling with two novel processes. First, the learn-to-express enables model depth scaling by allowing more important features to be captured by shallow layers. Second, the learn-to-compress allows embedding size scaling by employing information compression to compress embeddings' inner dependencies. The dual scalability of ESE enables it to be truncated into different scales, making it adaptable to varying computational resource requirements. Extensive experiments on STS and RAG have suggested that ESE can effectively produce high-quality embeddings with less model depth and embedding size, enhancing embedding inference efficiency.

ACKNOWLEDGEMENTS

Xianming Li and Jing Li's work has been supported by a grant from the Research Grants Council of the Hong Kong Special Administrative Region, China (Project No. PolyU/25200821), the Innovation and Technology Fund (Project No. PRP/047/22FX), and PolyU Internal Fund from RC-DSAI (Project No. 1-CE1E). Zongxi Li's work has been supported by Faculty Research Grants (SDS24A2) of Lingnan University, Hong Kong, and the Faculty Development Scheme (Project No. UGC/FDS16/E10/23), of Hong Kong Research Grants Council; Haoran Xie's work has been supported by the Faculty Research Grants (SDS24A8) and the Direct Grant (DR25E8) of Lingnan University, Hong Kong; Qing Li's work has been supported by Hong Kong Research Grants Council through Research Impact Fund (project no. R1015-23).

Here, we sincerely thank the reviewers and ACs for their valuable input, which has greatly improved our work.

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
