# OpenReview forum: "ESE: Espresso Sentence Embeddings"
_ICLR.cc/2025/Conference — ICLR 2025 Poster_

### Official Review · Reviewer_3a5X · 2024-10-28

**Soundness:** 3
**Presentation:** 3
**Contribution:** 2
**Rating:** 6
**Confidence:** 3

**Summary:**

This paper is concerned about sentence embeddings. This paper claims that previous optimizations on the efficiency of sentence embeddings largely lie in the embedding size yet ignore the model depth. However, both the model depth and the embedding size could have a large impact on the efficiency. So in this paper, espresso sentence embeddings are proposed to consider both two facets in a joint optimization framework including both learning-to-express and learning-to-compress. The learning-to-express attempts to compress the model depth while the learning-to-compress attempts to compress the embedding size. Experimental results show that espresso sentence embedding enjoy better performance-efficiency tradeoffs than compared baselines.

**Strengths:**

- The proposed espresso sentence embeddings are indeed the first of its kind in this area.
- The experimental results denote good performance-efficiency tradeoffs have been achieved.

**Weaknesses:**

- The idea, though not previous studied, is not that novel. And both two facets, i.e., the mode depth and embedding size, have been thoroughly studied in existing literature.

**Questions:**

- Why, in Figure 4a, spearman correlation is degraded along the increment of embedding size?

---

> ### Author Response · Authors · 2024-11-18
> **Response to Reviewer 3a5X**
>
> We would like to thank reviewer 3a5X for the constructive comments. Below are our responses to your concerns.
>
> ---
>
> ## Concern 1: The idea, though not previous studied, is not that novel. And both two facets, i.e., the mode depth and embedding size, have been thoroughly studied in existing literature.
>
> **Answer:**
>
> We greatly appreciate the reviewer‘s perspective on prior work in this area. **We would appreciate it if you could kindly point us to specific works that you find closely related, as this would help us better position our work and acknowledge relevant prior contributions.**
>
> Our contribution lies in the unique integration of:
>
> - Information compression directly into the training process through PCA on embedding dependencies
> - A unified framework that jointly optimizes both depth and size scaling for sentence embeddings
> - The ability to achieve flexible scaling with a single training iteration, unlike approaches that require multiple trainings for different sizes
>
> While individual aspects have been studied (e.g., embedding size in MRL), we believe the novelty comes from the specific technical approach using PCA during training, and the synergistic combination of depth and size scaling. Our approach can provide practical benefits of achieving both forms of scaling through a single-trained model.
>
> We will enhance our literature review to better contextualize our work within existing research. Please let us know if there are particular works we should discuss in more detail. Thank you so much!
>
> ---
>
>
> ## Concern 2: Why, in Figure 4a, spearman correlation is degraded along the increment of embedding size?
>
> **Answer:**
>
> We appreciate the reviewer's careful observation of the performance degradation with increasing embedding size in Figure 4a. Several factors can explain this interesting phenomenon at layer 16 (half-depth) of LLaMA2:
>
> The major reason might be attributed to the information flow in LLMs. In LLMs fine-tuned with LoRA, intermediate layers may not be optimized to produce meaningful sentence embeddings directly. Larger embedding sizes at intermediate layers may capture more "in-progress" features that haven't been fully refined for the sentence embedding task.
> The degradation suggests that using full embedding dimensions at intermediate layers may actually preserve noise or incomplete semantic information.
>
>
> We also have considerations over model architecture. Layer 16 represents a transition point in the model's feature-processing pipeline. The performance drop with larger dimensions suggests that the intermediate representations are still being transformed. Smaller dimensions may actually act as a beneficial information bottleneck, forcing the model to focus on the most essential semantic features.
>
> This observation reinforces the value of ESE's learn-to-compress process. ESE actively learns to distill the most relevant information into smaller dimensions, and the results demonstrate why naive dimension scaling at intermediate layers may not be optimal as our model still sees significant improvements against RAW representation and MRL embedding.
>
> We appreciate the reviewer’s sharp finding that suggests an interesting direction for future research into the relationship between embedding dimensionality and layer depth in large language models. We will acknowledge this point in our final version. Thank you!
>
>
> ---
>
> We hope Reviewer 3a5X could consider raising the score accordingly if we have addressed your concerns. If you have any further concerns, please don't hesitate to let us know.

---

### Official Review · Reviewer_XogJ · 2024-11-03

**Soundness:** 3
**Presentation:** 4
**Contribution:** 3
**Rating:** 6
**Confidence:** 4

**Summary:**

The paper proposes "Espresso Sentence Embedding" (ESE) for effective sentence embedding learning with two key processes "learn-to-express" and "learn-to-compress.” In the "learn-to-express" phase, the loss is computed using both the final layer and intermediate layer embeddings. In "learn-to-compress," ESE focuses on compressing embeddings by applying the softmax of the outer product of attention values. Experimental results demonstrate that ESE outperforms baselines like RAW (original model) and MRL (Matryoshka embedding; Kusupati et al., 2022) on STS and QA benchmarks. Furthermore, ablation studies confirm the robustness of ESE across various language models and embedding sizes.

**Strengths:**

**Motivation**

- The proposed work addresses the scalability challenges in sentence embedding that arise from fixed embedding sizes and a fixed number of layers. The training objective (learn-to-compress and learn-to-express) is well-motivated and effectively demonstrated throughout the paper.

**Methods and Experiments**

- The approach of directly training shallow layer embeddings (Section 3.2) significantly enhances embedding quality. Experimental results in Figure 3 show that even embeddings from shallow layers can achieve performance comparable to deeper layers in other models, highlighting the effectiveness of this approach.
- The novel compression method (Section 3.3) successfully captures interactions between vector dimensions, improving upon traditional PCA techniques, as shown in Table 2.

**Practicality**

- The framework presents a representation learning method that can be applied to low-resource scenarios.

**Writing**

- The paper's motivation is clearly written and easy to follow. The figures, particularly Figure 2, are well-designed and effectively illustrate the proposed methodology.

**Weaknesses:**

The experimental setup is generally well constructed, but I have little concerns about following issues:

1. Kusupati et al. (2022) conduct extensive experiments across various classification and retrieval tasks using the ImageNet-1K dataset. In contrast, the paper under review **restricts its experimental scope solely to STS tasks,** which raises **concerns about the generalizability of the proposed method.** If the authors intended to focus exclusively on the NLP domain, I recommend including experiments in areas where sentence embeddings are widely applicable, such as information retrieval (e.g. BEIR benchmarks[1]). Although the paper includes results from HotpotQA (Table 4), it omits key metrics for evaluating the quality of embeddings from the retriever, such as supporting facts EM and F1. As a result, it is challenging to evaluate the sole impact on the retriever (i.e., embedding) within the overall pipeline.
2. It is uncertain whether **comparing ESE’s intermediate layer performance with that of other models is entirely fair.** Obviously, these models are intended to use the final layer embedding for inference, so a comparison based on shallow-layer performance would be inappropriate. For practical significance, it would be better to **emphasize that “non-final-layer embeddings (e.g., <12th or <24th layer) of ESE outperform the final-layer embeddings (e.g., 12th or 24th layer) of MRL or RAW” ** This would highlight ESE’s potential time-saving benefits by reducing forward passes while guaranteeing the fair comparison. However, in the BGE-based results shown in Figure 3, ESE’s performance at layer 11 does not appear to surpass that of MRL or ESE’s own performance at layer 12, which raises concerns about claimed practical advantages of  ESE.

[1] Thakur, Nandan, et al. "Beir: A heterogenous benchmark for zero-shot evaluation of information retrieval models." arXiv preprint arXiv:2104.08663 (2021).

**Questions:**

1. Why do MRL and RAW show consistent results across compression sizes in Figure 5? Does compression not cause any performance change for these models?
2. In Section 3.3, the compression dimension $k$, to which the vector is reduced, appears to be fixed. However, compressed embedding sizes vary in Figures 3 and 4. How is this possible? Are these models using different compression dimensions at training and testing time, or are they trained with varying k values each time?

---

> ### Author Response · Authors · 2024-11-16
> **Response to Reviewer XogJ [1/3]**
>
> We would like to thank Reviewer XogJ for your constructive comments. Below are our answers to your concerns.
>
> ---
>
> ## Concern 1
> > Kusupati et al. (2022) conduct extensive experiments across various classification and retrieval tasks using the ImageNet-1K dataset. In contrast, the paper under review restricts its experimental scope solely to STS tasks, which raises concerns about the generalizability of the proposed method. If the authors intended to focus exclusively on the NLP domain, I recommend including experiments in areas where sentence embeddings are widely applicable, such as information retrieval (e.g. BEIR benchmarks[1]). Although the paper includes results from HotpotQA (Table 4), it omits key metrics for evaluating the quality of embeddings from the retriever, such as supporting facts EM and F1. As a result, it is challenging to evaluate the sole impact on the retriever (i.e., embedding) within the overall pipeline.
>
> **Answer:** We appreciate the reviewer's careful consideration of the information retrieval performance.
>
> Our work focuses on sentence embeddings. STS is a widely used benchmark for evaluating the quality of sentence embeddings [1-3]. Thus, the STS experiment serves as our main experiment. Other than that, we also test the end-to-end performance of RAG. Both experimental results demonstrate the scalability and effectiveness of the proposed ESE.
>
> **Following your advice, we add an extra experiment to evaluate the proposed ESE on information retrieval tasks.** We experiment with a subset of the BEIR benchmark since the evaluation of the full benchmark is not feasible due to time and computational resource limitations during the rebuttal stage. The results are presented **in Appendix Section B (in the latest revision of PDF)**. The experimental results demonstrate the effectiveness of our proposed ESE on information retrieval tasks.
>
>
> **Reference**
>
> - [1] Gao, T., Yao, X., & Chen, D. (2021). Simcse: Simple contrastive learning of sentence embeddings. arXiv preprint arXiv:2104.08821.
> - [2] Reimers, N. (2019). Sentence-BERT: Sentence Embeddings using Siamese BERT-Networks. arXiv preprint arXiv:1908.10084.
> - [3] Chuang, Y. S., Dangovski, R., Luo, H., Zhang, Y., Chang, S., Soljačić, M., ... & Glass, J. (2022). DiffCSE: Difference-based contrastive learning for sentence embeddings. arXiv preprint arXiv:2204.10298.

---

> ### Author Response · Authors · 2024-11-16
> **Response to Reviewer XogJ [2/3]**
>
> ## Concern 2
> > It is uncertain whether comparing ESE’s intermediate layer performance with that of other models is entirely fair. Obviously, these models are intended to use the final layer embedding for inference, so a comparison based on shallow-layer performance would be inappropriate. For practical significance, it would be better to **emphasize that “non-final-layer embeddings (e.g., <12th or <24th layer) of ESE outperform the final-layer embeddings (e.g., 12th or 24th layer) of MRL or RAW” ** This would highlight ESE’s potential time-saving benefits by reducing forward passes while guaranteeing the fair comparison. However, in the BGE-based results shown in Figure 3, ESE’s performance at layer 11 does not appear to surpass that of MRL or ESE’s own performance at layer 12, which raises concerns about claimed practical advantages of ESE.
>
> **Answer:**
>
> We appreciate the reviewer's careful consideration of the fairness in comparing intermediate layer performance. We would like to clarify our perspective:
>
> We hope to clarify the purpose and motivation of our work. While we agree that models are typically designed to use final layer embeddings, there are important real-world scenarios where computational resources constrain model usage.
>
> ESE specifically aims to improve intermediate layer representations for such resource-constrained settings where using full models may not be feasible. As demonstrated in Table 5, ESE's intermediate layers outperform the final layer embeddings of independently trained smaller models (e.g., ESE with BERTbase truncated to small scale achieves +0.35 improvement over BERTsmall, and +1.79 over BERTtiny when truncated to tiny scale).
>
> Regarding comparing with the final layer, we acknowledge the reviewer's suggestion that demonstrating shallow layers outperforming the final layer would be ideal. However, it's important to note that our baselines (BGE, UAE, Qwen) are already highly optimized and, in our experiments, all are fine-tuned on the target tasks. Given that ESE introduces no additional parameters (in fact, uses one less layer), surpassing such strong final layer performance for intermediate layers would be extremely challenging. **The value of ESE lies in making intermediate layers more usable when computational constraints prevent the use of full models, rather than outperforming state-of-the-art final layer performance.**
>
> We appreciate the reviewer’s very kind comment on practical benefits. We think ESE provides a flexible solution where users can choose appropriate intermediate layers based on their specific resource constraints while achieving better performance than equivalently sized standalone models. This enables more efficient resource utilization in production environments where computational budgets are limited.

---

> > ### Author Response · Authors · 2024-11-16
> > **Response to Reviewer XogJ [3/3]**
> >
> > ## Concern 3
> > > Why do MRL and RAW show consistent results across compression sizes in Figure 5? Does compression not cause any performance change for these models?
> >
> >
> > **Answer:**
> >
> > Thank you for raising this question. This figure intends to present how ESE performs with different compression sizes. Since only ESE employs PCA compression while other models, i.e. MRL and RAW, do not have a compression mechanism, we only adopt their performance with full-capacity embeddings, which are the best performance of both models, to show in the figure. We understand this figure may be somewhat misleading. We will also depict the performance of subembedding of MRL and RAW models in the Figure or provide clarification in the caption explicitly to avoid possible confusion.
> >
> > ---
> >
> > ## Concern 4
> > > In Section 3.3, the compression dimension, to which the vector is reduced, appears to be fixed. However, compressed embedding sizes vary in Figures 3 and 4. How is this possible? Are these models using different compression dimensions at training and testing time, or are they trained with varying k values each time?
> >
> >
> > **Answer:**
> >
> > Thank you for raising the question. **The quick answer is that they are not trained with varying k values each time, which is exactly what we are trying to avoid**.
> > Please allow us to clarify the relationship between training compression dimension and testing embedding sizes in this way:
> >
> > During the learn-to-compress training process, we use a fixed compression dimension, i.e. k=128 as mentioned in Section 3.3 to learn ESE, where PCA organizes features according to their importance, encouraging more salient information to be encoded in earlier dimensions. **Then Eq 6 aligns the first k dimension of the embedding with the first k dimension of their PCA output, namely "learn to compress"**.
> >
> > Through the learn-to-compress, **at the inference stage, we DO NOT do PCA anymore**. Instead, one can flexibly use any first-d dimension of the learned embedding (usually d>k). We have this flexibility because the learn-to-compress process makes the first-k dimension contain the most significant information.
> >
> > We would like to highlight that this is one of our major contributions, as the MRL needs to train the model with all varying embedding dimensions. We hope this can address your concern.
> >
> > ---
> >
> > **We hope Reviewer XogJ could consider raising the score accordingly if we have addressed your concerns. If you have any further concerns, please don't hesitate to let us know.**

---

> ### Comment · Reviewer_XogJ · 2024-11-23
> **Author response acknowledgement**
>
> I appreciate the authors’ responses to my comments and their clarifications. I consider the current score to be reasonable for its scope and novelty, and thus, I will retain my score as is.  I believe the provided clarifications could be incorporated into the next revision of the paper.  I would particularly recommend the following changes:
> - Figure 5 is misleading as it seems to indicate varying performance levels for different values of $k$. I suggest replacing the dashed lines with solid lines for MRL for RAW to better distinguish them from ESE with varying dimensions.
> Retrieval metrics (e.g., R@$k$, nDCG@$k$) for the RAG experiments on HotpotQA should be reported to better investigate the impact of the retrieval pipeline.
> The statement “For the ESE setup, the compression dimension k is set to 128 by default…” suggests that $k$ may vary during the training setup. I recommend clarifying that ESE is not trained with diverse $k$ values.
> - As the BEIR benchmark primarily highlights the performance gap between in-domain (MS MARCO) and out-of-domain datasets, an analysis of out-of-domain generalization would be beneficial after clarifying that ESE and other baselines are trained for MS MARCO.

---

> > ### Author Response · Authors · 2024-11-23
> > **Thank you!**
> >
> > **We sincerely thank the reviewer XogJ for their thorough engagement with our work and constructive suggestions. We are encouraged that the reviewer finds our responses satisfactory and considers a score of 6 appropriate for our contribution to scalable sentence embeddings.**
> >
> > We will incorporate all suggested clarifications in the final version:
> >
> > - We will revise Figure 5's visualization to better distinguish baselines from ESE
> > - We will add comprehensive retrieval metrics (R@K, nDCG@K) for HotpotQA experiments
> > - Clarification on ESE using a fixed k during training will be added
> > - Include a comprehensive analysis of the BEIR benchmark
> >
> > Thank you again!

---

### Official Review · Reviewer_876W · 2024-11-03

**Soundness:** 3
**Presentation:** 2
**Contribution:** 2
**Rating:** 5
**Confidence:** 5

**Summary:**

This paper follows the motivation of Matryoshka Representation Learning (MRL) to learn sentence representation at different granularities, allowing a single embedding to adapt to the computational constraints of downstream tasks. The main idea is to use the PCA compressed representation to supervise representation learning at each grain. They show some positive results on STS tasks, more specifically, compared to previous works, their method makes the embedding extracted in the inner layers better, which is aligned with their design. Also, some ablation studies are provided to support the plausibility of their design. Basically, the paper is clear and well-written. However, I hold some concerns with their motivation and design, see the weaknesses part.

**Strengths:**

1. The paper is well-written, showing clearly the gains from the inner layers upon raw and MRL design.
2. I appreciate their experiments with different model scales and compression strategies, which clearly demonstrate the ablation and provide realistic comparison results.

**Weaknesses:**

1) The design is complex: there are a lot of hyperparameters to tune with, like the weight for Loss_le, Loss_lc, the combination of MSE and KL-div, and joint training. It is hard to provide insights into where the main gain comes from.

2) Is embedding dimension k also a hyperparameter? If so, does this mean that achieving different types or lengths of compressed representations requires multiple training iterations?

3) I doubt real-world applications for this method. Their main motivation is to reduce inference latency, where the most straightforward way is to use a smaller backbone. However, as shown in Table 5, the gain compared with BERT_small is marginal.

**Questions:**

1) Is embedding dimension k also a hyperparameter? If so, does this mean that achieving different types or lengths of compressed representations requires multiple training iterations?

2) Does "PCA on sentence embeddings" in Table 2 mean directly applying PCA on the high-dim representations? If so, from the storage angle, the advantage of the proposed method is limited?

---

> ### Author Response · Authors · 2024-11-21
> **Response to Reviewer 876W [1/2]**
>
> We would like to thank Reviewer 876W for your helpful comments. Below are our answers to your concerns.
>
> ---
>
> ### Concern 1: The design is complex: there are a lot of hyperparameters to tune with, like the weight for Loss_le, Loss_lc, the combination of MSE and KL-div, and joint training. It is hard to provide insights into where the main gain comes from.
>
> **Answer:**
>
> Thank you for raising the question about design complexity. We would like to emphasize that our model design is straightforward and builds upon well-established techniques and aims to simplify the overall approach.
>
> The joint training strategy we employ is widely adopted practice in the field. particularly successful and widely recognized in mixture-of-experts models [3, 4]. Similarly, our use of combined MSE and KL divergence losses is grounded in prior research demonstrating their complementary benefits in knowledge distillation [1, 2].
>
> We structure our approach into two clear phases, i.e. learn-to-express and learn-to-compress. The learn-to-express is designed to enhance layer representations, and the learn-to-compress is designed to support scalability. The whole process is intuitive, and each component serves a distinct purpose and uses established loss functions. The mentioned two losses, i.e. loss_le and loss_lc, are necessary for both phases. Although we set two hyperparameters \alpha and \beta, which are common approaches in joint training, they largely follow natural choices, e.g. equal weighting \alpha=\beta=1, for joint training.
>
> Through our ablation studies and empirical analysis, we found that this design offers a practical balance between complexity and effectiveness. Each component contributes meaningfully to the overall performance, as demonstrated in our experimental results. Hope you can find our response satisfactory. Please feel free to let us know if you have any further comments.
>
> **Reference:**
>
> [1] Kim, T., Oh, J., Kim, N., Cho, S., & Yun, S. Y. (2021). Comparing kullback-leibler divergence and mean squared error loss in knowledge distillation. IJCAI 2021. https://www.ijcai.org/proceedings/2021/0362.pdf
>
> [2] Cui, J., Tian, Z., Zhong, Z., Qi, X., Yu, B., & Zhang, H. (2023). Decoupled kullback-leibler divergence loss. NeurIPS 2024. https://neurips.cc/virtual/2024/poster/94462
>
> [3] Dai, D., Deng, C., Zhao, C., Xu, R. X., Gao, H., Chen, D., ... & Liang, W. (2024). Deepseekmoe: Towards ultimate expert specialization in mixture-of-experts language models. ACL 2024. https://aclanthology.org/2024.acl-long.70/
>
> [4] He, S., Fan, R. Z., Ding, L., Shen, L., Zhou, T., & Tao, D. (2023). Merging experts into one: Improving computational efficiency of mixture of experts. EMNLP 2023. https://aclanthology.org/2023.emnlp-main.907/
>
> ---
>
> ### Concern 2: Is embedding dimension k also a hyperparameter? If so, does this mean that achieving different types or lengths of compressed representations requires multiple training iterations?
>
> **Answer:**
>
> Thank you so much for raising this question. We would like to clarify that k is a hyperparameter controlling how many earlier dimensions we try to compress the information into. However, different from MRL, which needs to train multiple embedding dimensions, our ESE only needs to train the model with a specific k for only once. The learn-to-compress process naturally orders features by importance through PCA and compresses the most significant information in the earlier dimensions. In the inference stage, we **DO NOT** do PCA anymore. One can flexibly use the first-d dimension (usually d>k) for inference. This approach allows us to choose any embedding size at inference with maximum information preserved without training multiple models. We have highlighted this part **in Lines 222-226**. Thank you so much!

---

> > ### Author Response · Authors · 2024-11-21
> > **Response to Reviewer 876W [2/2]**
> >
> > We would like to thank Reviewer 876W for your helpful comments. Below are our answers to your concerns.
> >
> > ---
> >
> > ### Concern 3: I doubt real-world applications for this method. Their main motivation is to reduce inference latency, where the most straightforward way is to use a smaller backbone. However, as shown in Table 5, the gain compared with BERT_small is marginal.
> >
> > **Answer:**
> >
> > Thank you for this important question about the practical utility of ESE. While using a smaller backbone is indeed straightforward, we believe ESE offers several important practical advantages that go beyond simple performance metrics.
> > A key benefit of ESE is its ability to enable flexible deployment from a single model. Unlike the conventional approach of training separate small models, ESE enables dynamic adaptation to varying resource constraints without maintaining multiple models. This significantly reduces storage and deployment complexity, which is particularly valuable in production environments where resource availability fluctuates.
> > Regarding performance, while the improvement over BERTsmall (+0.35) may appear marginal in isolation, it's important to consider the broader context. These gains are achieved without any additional training or model maintenance costs, they come "for free" in terms of infrastructure.
> >
> > We would like to further highlight the meaning of **Cost Efficiency** and **Real-world Application**. Training and maintaining multiple backbone models is expensive in terms of computational resources, storage, and operational complexity. ESE addresses these challenges by providing a single-model solution that can adapt to different requirements.
> > In terms of real-world benefits, ESE allows dynamic adjustment without model switching or redeployment to handle varying load conditions and operate across different hardware configurations in the production environments.
> > These practical advantages, i.e. flexible scaling, reduced maintenance overhead, and dynamic adaptation capabilities, make ESE particularly valuable in real production environments, where operational efficiency is often as crucial as raw performance metrics.
> > We appreciate the opportunity to clarify these points and hope our work can contribute to the ongoing discussion about efficient model deployment in resource-constrained environments. Thank you.
> >
> >
> > ### Concern 4: Does "PCA on sentence embeddings" in Table 2 mean directly applying PCA on the high-dim representations? If so, from the storage angle, the advantage of the proposed method is limited?
> >
> > Thank you for this important question. We think there may be a misunderstanding point in the comment and would like to clarify the ablation experiments in Table 2.
> >
> > As discussed **in Lines 373-375**, our proposed method applies PCA on embedding dependencies (A_i^d in Eq. 3-5), not directly on sentence embeddings. This novel approach represents a key technical contribution of our work and, as shown in Table 2, achieves better performance than direct PCA on embeddings. The "PCA on sentence embeddings" entry in Table 2 represents an ablative design we tested for comparison, where PCA is applied directly to embeddings (X_i^d) following traditional approaches.  We include it specifically in our ablation study to demonstrate why our dependency-based approach is more effective. It's important to note that this is just an experimental ablation setting, not our default model.
> >
> > We also want to emphasize the critical distinction between the training and inference phases in our approach. Our learn-to-compress process is designed to only utilize PCA during training to guide feature organization. During inference, we can directly use the top-k dimensions without any PCA computation, eliminating computational overhead while maintaining the benefits of compression-guided feature organization. This design choice is fundamental to achieving efficient deployment.
> >
> >
> > ---
> >
> > We hope Reviewer 876W could consider raising the score accordingly if we have addressed your concerns. If you have any further concerns, please don't hesitate to let us know. We look forward to your further comments!

---

> > > ### Author Response · Authors · 2024-11-26
> > > **Look forward to your reply!**
> > >
> > > We understand Reviewer 876W might still have concerns regarding the real-world applications of the proposed ESE.
> > > Please allow us to use an example to illustrate ESE's broader value proposition through a concrete business scenario that may better address the reviewer's concerns about real-world applicability. Let’s consider a practical scenario:
> > >
> > > A company deploys LM-based services across diverse applications. Some applications, such as real-time content search and retrieval, require quick responses and thus need small, efficient models. Meanwhile, more sophisticated applications like customer service chatbots require deeper semantic understanding and larger models. The company must also account for varying device capabilities across mobile applications and edge devices, each with its own resource constraints.
> > >
> > > Traditional approaches need to train and maintain separate models for each scenario, which poses significant challenges. When knowledge needs updating, security vulnerabilities need patching, or business requirements change, complete retraining is required for all models. This creates a complex deployment pipeline with substantial operational overhead.
> > >
> > > ESE addresses these practical challenges by providing a single trained model that dynamically scales, enabling flexible deployment across different computational budgets. This unified approach simplifies model management and updates, allowing easy adaptation to new resource constraints. The benefits become particularly significant when working with larger models like LLaMA or Mistral, where the cost and complexity of maintaining multiple differently-sized versions would be substantial in terms of both storage and operational overhead.
> > >
> > > We appreciate Reviewer 876W’s time for the review. We look forward to Reviewer 876W’s reply to let us know whether our responses have addressed your concerns. Thank you!

---

> > > > ### Author Response · Authors · 2024-12-01
> > > > **Look forward to your reply!**
> > > >
> > > > Dear Reviewer 876W,
> > > >
> > > > Thank you again for your valuable comments on our work. **As the rebuttal deadline approaches, we would greatly appreciate hearing from you regarding whether our responses have adequately addressed their concerns.**
> > > >
> > > > We have responded to your concerns regarding model design and real-world applications. In particular, we have further elaborated real-world applications of our ESE with an example as support.
> > > >
> > > > | Topic | Response Link |
> > > > |-------|---------------|
> > > > | Model Design (Part 1) | [Response](https://openreview.net/forum?id=plgLA2YBLH&noteId=rTRAsIZ2cH) |
> > > > | Model Design (Part 2) | [Response](https://openreview.net/forum?id=plgLA2YBLH&noteId=x7Q5rzEYUn) |
> > > > | Real-world Application Example | [Response](https://openreview.net/forum?id=plgLA2YBLH&noteId=I9mBbBOfMy) |
> > > >
> > > > Your continued engagement in this discussion is crucial for further improving our work. We welcome any additional feedback or suggestions you may have. Thank you so much!
> > > >
> > > > Best,
> > > >
> > > > Authors.

---

### Official Review · Reviewer_527o · 2024-11-03

**Soundness:** 3
**Presentation:** 2
**Contribution:** 3
**Rating:** 6
**Confidence:** 4

**Summary:**

This work introduces a method for fine-tuning sentence embeddings while simultaneously providing flexibility in model size and embedding size. The method consists of two parts: learn-to-express, which trains different model depths, and learn-to-compress, which focuses on compressing information into a sub-embedding. The authors provide extensive experimental results on STS datasets and RAG on QA, including visualizations for the quality of the embeddings as well as results on inference efficiency. For the ablation studies, they investigate PCA on different dependencies, the effect of embedding sizes for PCA, and weighting present in the loss function.

**Strengths:**

1. The authors fuse two existing ideas (model truncation and embedding truncation) into a single training process, which is a somewhat limited novelty. However, this provides much-needed flexibility for real-world use cases.
2. The method is backed up by empirical results on seven STS datasets, one RAG dataset, and ablation studies that justify method design.
3. The paper uses figures to better explain concepts, which is nice. However, I often found it lacking details for experiments and ablation studies.
4. This paper introduces the idea of training multiple configurations of a model at once. This is very useful for efficiency-critical and memory-constrained environments. The method can be handy for engineering use cases.

**Weaknesses:**

## 1. Unfocused contribution statement

> To the best of our knowledge, we are the first to learn sentence embedding with information compression, presenting scalable embedding inference to both model depths and embedding dimensions

> We are the first to employ the information compression technique to scale sentence embeddings"

If by "information compression" the authors are referring to model depth, there are already existing papers on this topic, some of which are even cited (e.g., "BeLLM: Backward Dependency Enhanced Large Language Model for Sentence Embeddings" and "Are ELECTRA's Sentence Embeddings Beyond Repair? The Case of Semantic Textual Similarity"). Additionally, other cited works focus on reducing the embedding size (e.g., "Matryoshka Representation Learning"). Please clarify your wording and specify precisely how your method differs from these approaches.

## 2. Incomplete related work

The related work section is quite incomplete, as it includes only a few well-known references, most of which are not closely connected to the paper's topic. For example, a few papers that address closely related topics (excluding the aforementioned references) are:

- Interpreting Pretrained Contextualized Representations via Reductions to Static Embeddings
- WhiteningBERT: An Easy Unsupervised Sentence Embedding Approach
- What does BERT learn about the structure of language?
- How Contextual are Contextualized Word Representations? Comparing the Geometry of BERT, ELMo, and GPT-2 Embeddings
- BERT Has More to Offer: BERT Layers Combination Yields Better Sentence Embeddings

## 3. Unmotivated design choices

The paper is very empirical in nature, which is not a bad thing. While most of the method's design is supported by intuitive explanations, I find the rationale for choosing the "align" loss function insufficiently explained. Ideally, a more theoretical approach and justification for introducing the loss functions would significantly strengthen the paper.

> Specifically, we use Euclidean distance, i.e., IndexFlatL2 in faiss, as the similarity measurement

Why use the Euclidean distance if you are optimizing for cosine similarity?

> Note that we leave the last layer unweighted due to its critical role in capturing sentence-level semantics

The reference paper mentioned only tests this on LLMs, so what is the design justification for using it on MLMs?

## 4. Potentially incorrect/misleading conclusions

> It allows the direct use of the first k-dimensional sub-embeddings at inference without repeatedly performing PCA and improves inference efficiency

I am not convinced by the second part of this statement. I can see that this improves efficiency ONLY if you consider using PCA. If you don't use any compression, the efficiency would be roughly the same, wouldn't it?

Furthermore, Figures 3, 4, and ≺ Avg. columns for Tables 1 and 2 provide an unfair comparison -- RAW and MRL do not tune the inner transformer layers. This comparison is repeated a couple of times throughout the paper as an insight:

> RAW and MRL exhibit inferior performance in the shallow layers

> This shows the effectiveness of ESE's learn-to-express process in encoding the more important information into shallow layers

> Notably, ESE achieves a significant improvement at shallow layers, demonstrating its superior scalability

This leads an unwary reader to an incorrect conclusion. It would make more sense to tune each layer separately and compare it with your method.

Furthermore, Table 5 is a somewhat unfair comparison. Truncating the model depth does not mean the model has the same number of parameters as BERT_smal or BERT_tiny (due to different hidden dimensions). I would advise the authors to add the information on number of parameters and mention this while commenting on the results.

The comparison of distances after using TSNE is debatable. TSNE preserves the global structure, but measuring distances after TSNE does not make sense (cf. [https://datascience.stackexchange.com/questions/17314/are-t-sne-dimensions-meaningful](https://datascience.stackexchange.com/questions/17314/are-t-sne-dimensions-meaningful)). For more rigor, I would instead use MDS for this visualization and comparison.

## Minor remarks

- Figure 7 seems to lack details on which layer was used to create the embeddings on the subfigures and which 128 dimensions were chosen for the RAW method.

- For Table 1, I would advise to put the links into the footnotes.

- I know you use CLS pooling in your experiments, but the implementation for mean pooling is incorrect.

```
    if pooling_strategy == 'avg':
        outputs = torch.sum(
            outputs * inputs["attention_mask"][:, :, None], dim=1) / torch.sum(inputs["attention_mask"])
```

- Is Equation 6 missing a sum over k?

**Questions:**

## Q1:

> The possible reason is the ESE's high-quality embeddings in varying scaled sizes allow easier and more effective indexing, thus improving RAG efficiency

I do not see how this affects the indexing process. Can you elaborate a bit more?

## Q2:

Could you share the code for the inference used in the case of MRL or point me to the part of the code that does this in the anonymized repository?

---

> ### Author Response · Authors · 2024-11-16
> **Response to Reviewer 527o | Part 1**
>
> We would like to thank reviewer 527o for his thorough review and constructive comments. Below are our responses to your concerns.
>
> ---
>
> ## Concern 1: Unfocused contribution statement
>
> >  To the best of our knowledge, we are the first to learn sentence embedding with information compression, presenting scalable embedding inference to both model depths and embedding dimensions
>
> > We are the first to employ the information compression technique to scale sentence embeddings
>
> > If by "information compression" the authors are referring to model depth, there are already existing papers on this topic, some of which are even cited (e.g., "BeLLM: Backward Dependency Enhanced Large Language Model for Sentence Embeddings" and "Are ELECTRA's Sentence Embeddings Beyond Repair? The Case of Semantic Textual Similarity"). Additionally, other cited works focus on reducing the embedding size (e.g., "Matryoshka Representation Learning"). Please clarify your wording and specify precisely how your method differs from these approaches.
>
> **Answer:**
>
> Thank you so much for this important feedback about precision in our novelty claims and for pointing out some related works! We understand that our current wording may require clarification to better distinguish our contributions from prior work.
>
> Please allow us to clarify more precisely. We are the first to integrate PCA-based compression directly into the sentence embedding training process. Both BeLLM and ELECTRA Focus on architecture modifications or loss functions and do not employ information compression techniques. Although MRL adopts size reduction, they train multiple embedding heads for different sizes and do not **compress** information during training.
>
> ESE uniquely compresses embedding inner dependencies (not just dimensions and incorporates compression as part of the training objective, enabling dual scalability in both model depth and embedding size.
>
> We introduce a learn-to-compress process that:
> - Uses PCA to organize features by importance during training
> - Aligns truncated embeddings with compressed representations
> - Enables flexible scaling without multiple training iterations
>
> We will revise our novelty claim to more precisely highlight these specific technical contributions and better differentiate our approach from existing work.
>
>
> ## Concern 2: Incomplete related work
>
> > The related work section is quite incomplete, as it includes only a few well-known references, most of which are not closely connected to the paper's topic. For example, a few papers that address closely related topics (excluding the aforementioned references) are:
> > - Interpreting Pretrained Contextualized Representations via Reductions to Static Embeddings
> > - WhiteningBERT: An Easy Unsupervised Sentence Embedding Approach
> > - What does BERT learn about the structure of language?
> > - How Contextual are Contextualized Word Representations? Comparing the Geometry of BERT, ELMo, and GPT-2 Embeddings
> > - BERT Has More to Offer: BERT Layers Combination Yields Better Sentence Embeddings
>
>
> **Answer:**
>
> Thank you so much for raising this issue and for your expert knowledge in the field. We will include and discuss the suggested references in our paper to strengthen our literature review in the camera-ready version. Please feel free to let us know if we miss any related work.

---

> > ### Author Response · Authors · 2024-11-16
> > **Response to Reviewer 527o | Part 2**
> >
> > ## Concern 3: Unmotivated design choices
> >
> > ### 1) Q1: Why use the Euclidean distance if you are optimizing for cosine similarity?
> > > The paper is very empirical in nature, which is not a bad thing. While most of the method's design is supported by intuitive explanations, I find the rationale for choosing the "align" loss function insufficiently explained. Ideally, a more theoretical approach and justification for introducing the loss functions would significantly strengthen the paper.
> > Specifically, we use Euclidean distance, i.e., IndexFlatL2 in faiss, as the similarity measurement
> >
> > **Answer:**
> > Thank you for raising this issue. RAG is a downstream application. The part concerned aims to demonstrate the capabilities of the proposed ESE in a real-world application. In RAG, our experimental setup generally follows common settings [1].
> >
> > Cosine similarity is particularly related to L2 distance. For L2-normalized vector A and B, $||A||^2 = 1$, $||B||^2 = 1$, their L2 distance is computed as follows:
> >
> > $$||A-B||^2 = (A - B) \cdot (A - B) = ||A||^2 + ||B||^2 - 2(A \cdot B) = 2(1 - cos(A, B))$$
> >
> > This equation is from [2]. We can see that the cosine similarity is actually part of the L2 distance. Therefore, theoretically speaking, we do not believe using L2 is a big issue. The difference in actual use is that cosine similarity is the higher, the better, while L2 distance is the lower, the better.
> >
> > **Reference:**
> >
> > - [1] Douze, M., Guzhva, A., Deng, C., Johnson, J., Szilvasy, G., Mazaré, P. E., ... & Jégou, H. (2024). The faiss library. arXiv preprint arXiv:2401.08281.
> > - [2] https://en.wikipedia.org/wiki/Cosine_similarity#Angular_similarity
> >
> > ---
> >
> > ### 2) Q2: The reference paper mentioned only tests this on LLMs, so what is the design justification for using it on MLMs?
> > > Note that we leave the last layer unweighted due to its critical role in capturing sentence-level semantics
> >
> > **Answer:**
> >
> > Thank you for this careful observation about the referenced design choice. You are correct that our cited work (Li & Li, 2024b) only validates this design choice for LLMs. We acknowledge that it lacks rigorous theoretical justification. We referenced this work primarily in designing the weighting mechanism, and our subsequent empirical results corroborate their findings. Specifically, we noticed that the RAW model’s last layer significantly outperforms shallow layers in Figure 3, showing that it is the most powerful layer in the pretrained model. To preserve the original expressiveness of the last layer after finetuning, we decided not to apply a weight term to the last layer.
> >
> > To avoid any confusion, we have reworded the sentence as ”Our empirical findings show that, in the RAW model, the last layer significantly outperforms shallow layers, which agrees with the findings in Li & Li (2024b) studying layers in LLMs. Therefore, we leave the last layer unweighted to preserve its critical role during training”. Please kindly let us know whether we have adequately addressed your concern. Thank you again.

---

> > > ### Author Response · Authors · 2024-11-16
> > > **Response to Reviewer 527o | Part 3 [1/2]**
> > >
> > > ## Concern 4: Potentially incorrect/misleading conclusions
> > >
> > > ### 1) Q1: I am not convinced by the second part of this statement. I can see that this improves efficiency ONLY if you consider using PCA. If you don't use any compression, the efficiency would be roughly the same, wouldn't it?
> > > > It allows the direct use of the first k-dimensional sub-embeddings at inference without repeatedly performing PCA and improves inference efficiency
> > >
> > >
> > > **Answer:**
> > >
> > > Thank you for your very careful observation about this claim. You make a valid point that requires clarification. You are right that our statement about improved efficiency is only relative to using PCA at inference time. We did not intend to compare efficiency against uncompressed models because the efficiency is indeed roughly the same. We will revise the statement more precisely: “It allows the direct use of the first k-dimensional sub-embeddings at inference without the computational overhead of PCA while maintaining the benefits of compression-guided feature organization.” Thank you for helping us identify this issue. The revised version will more accurately represent the efficiency aspects of our method.
> > >
> > >
> > > ---
> > >
> > > ### 2) Q2: This leads an unwary reader to an incorrect conclusion. It would make more sense to tune each layer separately and compare it with your method.
> > >
> > > > Furthermore, Figures 3, 4, and ≺ Avg. columns for Tables 1 and 2 provide an unfair comparison -- RAW and MRL do not tune the inner transformer layers. This comparison is repeated a couple of times throughout the paper as an insight:
> > > RAW and MRL exhibit inferior performance in the shallow layers
> > > This shows the effectiveness of ESE's learn-to-express process in encoding the more important information into shallow layers
> > > Notably, ESE achieves a significant improvement at shallow layers, demonstrating its superior scalability
> > >
> > >
> > > **Answer:**
> > >
> > > We appreciate the reviewer’s concern about comparison fairness. However, we respectfully disagree with the comparison of independently tuning each layer, as it would actually contradict the paper’s core efficiency goals.
> > >
> > > The core motivation of ESE is to improve inference efficiency through a unified model that can scale across different depths and dimensions. This provides essential practicality in realistic scenarios where practitioners are able to dynamically adjust model depth based on computational resources by maintaining a single deployable model only.
> > >
> > > Although we haven’t tried it, we also hypothesize that it is possible to achieve some improvements by independently training on each layer. While we acknowledge that independently training each layer might achieve some improvements, as this would allow each layer to be fully optimized for the task without needing to propagate information to deeper layers, such an approach would fundamentally contradict the efficiency goals by requiring N separate models for N different layers, which multiplies storage requirements by N.
> > >
> > > Additionally, this may further lead to the implication of not only independently training each layer but also independently training on each possible dimension, which will be computationally prohibitive in practice, albeit possibly effective. We try to achieve a balance between model performance and training complexity.
> > >
> > > We believe our current comparison methodology appropriately evaluates ESE against existing approaches within the context of our efficiency-focused goals. The observed performance improvements in shallow layers demonstrate ESE's effectiveness at learning a unified, scalable representation rather than requiring separate models for each depth configuration.

---

> > > > ### Author Response · Authors · 2024-11-16
> > > > **Response to Reviewer 527o | Part 3 [2/2]**
> > > >
> > > > ## Concern 4: Potentially incorrect/misleading conclusions
> > > >
> > > > ### 3) Q3: Furthermore, Table 5 is a somewhat unfair comparison. Truncating the model depth does not mean the model has the same number of parameters as BERTsmall or BERTtiny (due to different hidden dimensions). I would advise the authors to add the information on number of parameters and mention this while commenting on the results.
> > > >
> > > >
> > > > **Answer:**
> > > >
> > > > We acknowledge that our scaled-down versions maintain the wider architecture of BERTbase and thus have more parameters than the independently trained BERTsmall and BERTtiny models.
> > > >
> > > > We will revise Table 5 to clarify this difference and discuss the potential issues. However, we would like to emphasize that the primary goal of this comparison is to demonstrate ESE's scalability capabilities. Our key contribution is enabling the dynamic scaling of a single model, and the comparison showcases that a larger model can be effectively scaled down to arbitrary depths while maintaining the competitive performance of sub-embeddings. This offers practical flexibility that separate specialized models cannot provide.
> > > >
> > > > If the reviewer considers the comparison is not suitable, we can consider removing this discussion.
> > > >
> > > > ---
> > > >
> > > >
> > > > **We sincerely thank the reviewer for all the constructive comments on our work. The reviewer’s expertise and insight are highly appreciated. Please kindly give us some time to prepare for the rest of the questions, we will get back to you as soon as we can. Thank you so much and have a nice weekend!**

---

> > > > > ### Author Response · Authors · 2024-11-17
> > > > > **Response to Reviewer 527o | Part 4**
> > > > >
> > > > > Dear Reviewer 527o,
> > > > >
> > > > > Here is the final part of our answer to your concerns and questions.
> > > > >
> > > > >
> > > > > ## Concern 5: Figure 7 seems to lack details on which layer was used to create the embeddings on the subfigures and which 128 dimensions were chosen for the RAW method.
> > > > >
> > > > >
> > > > > **Answer:** Thank you for raising the question. We evaluate the compression quality in Figure 7, the selected layer is the last layer for RAW, MRL, and ESE. We will revise the caption of Figure 7 to make it clearer.
> > > > >
> > > > > ---
> > > > >
> > > > > ## Concern 6: For Table 1, I would advise to put the links into the footnotes.
> > > > >
> > > > > **Answer:**
> > > > >
> > > > > Thanks for your advice on the layout, we will put links in the footnote in the camera-ready version.
> > > > >
> > > > > ---
> > > > >
> > > > > ## Concern 7: I know you use CLS pooling in your experiments, but the implementation for mean pooling is incorrect.
> > > > >
> > > > > **Answer:**
> > > > >
> > > > > Thank you for pointing out the bug in mean pooling. We have fixed it in the latest code. Since we use CLS pooling, it does not affect the results.
> > > > >
> > > > >
> > > > > ## Concern 8: Is Equation 6 missing a sum over k?
> > > > >
> > > > >
> > > > > **Answer:**
> > > > >
> > > > > Thank you for raising this issue. **There is no need to sum over k since k is a hyperparameter. We do not train with various k each time, which is exactly what we are trying to avoid.**
> > > > >
> > > > > Please allow us to clarify the relationship between training compression dimension and testing embedding sizes in this way:
> > > > > During the learn-to-compress training process, we use a fixed compression dimension, i.e. k=128 as mentioned in Section 3.3 to learn ESE, where PCA organizes features according to their importance, encouraging more salient information to be encoded in earlier dimensions. Then Eq 6 aligns the first k dimension of the embedding with the first k dimension of their PCA output.
> > > > >
> > > > > At the inference stage, we DO NOT do PCA anymore. Instead, one can flexibly use any first-d dimension of the learned embedding (usually d>k). We have this flexibility because the learn-to-compress process makes the first-k dimension contain the most significant information.
> > > > >
> > > > > ---
> > > > >
> > > > > ## Concern 9: I do not see how this affects the indexing process. Can you elaborate a bit more?
> > > > > > The possible reason is the ESE's high-quality embeddings in varying scaled sizes allow easier and more effective indexing, thus improving RAG efficiency
> > > > >
> > > > > **Answer:**
> > > > >
> > > > > ESE can directly impact vector indexing efficiency. When using vector indexing tools like faiss, the embedding dimension affects both storage and index complexity. ESE allows users to flexibly choose smaller embedding dimensions while maintaining high quality, reducing both storage overhead and indexing speed. When handling large data, users can also truncate the layers of ESE model to accelerate the encoding phase.
> > > > >
> > > > > ---
> > > > >
> > > > > ## Concern 10: Could you share the code for the inference used in the case of MRL or point me to the part of the code that does this in the anonymized repository?
> > > > >
> > > > >
> > > > > **Answer:**
> > > > >
> > > > > Sure, absolutely no problem. We have uploaded the NLI inference code called eval_nli.py, you can find it in the anonymized repository on the first page (sorry that we cannot paste the link here to avoid violating the rebuttal requirement). For MRL inference, you can truncate the embedding by specifying the `–embedding_size`.
> > > > >
> > > > > ---
> > > > >
> > > > > **We sincerely thank reviewer 527o again for the thorough review and all the constructive comments on our work.**
> > > > >
> > > > > **We hope Reviewer 527o could consider raising the score accordingly if we have addressed your concerns. If you have any further concerns, please don't hesitate to let us know. We look forward to your further comments!**
> > > > >
> > > > >
> > > > > Best,
> > > > >
> > > > > Authors.

---

> > > > > > ### Comment · Reviewer_527o · 2024-11-20
> > > > > > **Reply to replies**
> > > > > >
> > > > > > > This equation is from [2]. We can see that the cosine similarity is actually part of the L2 distance. Therefore, theoretically speaking, we do not believe using L2 is a big issue. The difference in actual use is that cosine similarity is the higher, the better, while L2 distance is the lower, the better.
> > > > > >
> > > > > > Thank you for the clarification, as I was not aware that the embeddings you use are already normalized. I would advise you to mention this detail in the paper as this ensures the nearest neighbors are the same regardless of using L2 or cosine.
> > > > > >
> > > > > > > Specifically, we noticed that the RAW model's last layer significantly outperforms shallow layers in Figure 3, showing that it is the most powerful layer in the pretrained model. To preserve the original expressiveness of the last layer after finetuning, we decided not to apply a weight term to the last layer.
> > > > > >
> > > > > > Isn't this expected since in the RAW mode you're using the output of the final layer for fine-tuning? The intuition is that if you fine-tune any layer it's going to achieve higher scores than other layers, isn't it?
> > > > > >
> > > > > > > independently training each layer but also independently training on each possible dimension, which will be computationally prohibitive in practice, albeit possibly effective
> > > > > >
> > > > > > I am aware of this approach being extremely computationally inefficient, and I would just like to say that I do not propose it for the method comparison.
> > > > > >
> > > > > > > such an approach would fundamentally contradict the efficiency goals by requiring N separate models for N different layers, which multiplies storage requirements by N.
> > > > > >
> > > > > > I do not fully agree with your statement. This approach ONLY contradicts training efficiency, while inference efficiency (for the model, not the index) isn't affected. I don't think the storage requirement is a valid point, as you can easily overcome this by finding out how big of a model you can use for your use case, and only then fine-tune models up to the allowed size. Besides, you can only save the best model in each of these iterations, requiring only storage for a single model, instead of N. However, I do agree that using your method you can achieve training and inference efficiency at the same time, and you don't need to be aware of the possible use case model sizes ahead.
> > > > > >
> > > > > > Still, I do not think your arguments resolve the crux of the issue. Your method tunes inner layers directly, while RAW and MRL do not. I have a feeling this is actually comparing probing and fine-tuning. Can you explain why would we even expect RAW or MRL to achieve high-scoring inner representations if they are not directly tuned?
> > > > > >
> > > > > > > There is no need to sum over k since k is a hyperparameter. We do not train with various k each time, which is exactly what we are trying to avoid.
> > > > > >
> > > > > > Thank you for this clarification, initially I missed this detail.
> > > > > >
> > > > > > > ESE can directly impact vector indexing efficiency. When using vector indexing tools like faiss, the embedding dimension affects both storage and index complexity. ESE allows users to flexibly choose smaller embedding dimensions while maintaining high quality, reducing both storage overhead and indexing speed. When handling large data, users can also truncate the layers of ESE model to accelerate the encoding phase.
> > > > > >
> > > > > > Isn't this vector indexing efficiency already achieved by MRL? However, I am aware that your method is more efficient in the model inference phase, as MRL does not provide a mechanism for layer truncation.
> > > > > >
> > > > > > > Sure, absolutely no problem. We have uploaded the NLI inference code called eval_nli.py, you can find it in the anonymized repository on the first page (sorry that we cannot paste the link here to avoid violating the rebuttal requirement). For MRL inference, you can truncate the embedding by specifying the –embedding_size.
> > > > > >
> > > > > > Originally, I was most interested in how inference affects Figure 6a when I first asked this question, however this does not clarify much for me. I'm confused about the time taken for MRL for different layer indexes. If you're using MRL for the training regime, doing inference up to a certain layer, and finally slicing the embedding up to a certain dimension, shouldn't the inference times be roughly the same for MRL and ESE? The current plot for MRL looks like you're doing inference using the whole model, and only then choosing a certain layer and slicing the embedding.

---

> > > > > > > ### Author Response · Authors · 2024-11-21
> > > > > > > **Reply to the further comments of Reviewer 527o | Part 1**
> > > > > > >
> > > > > > > Dear Reviewer 527o,
> > > > > > >
> > > > > > > Thank you very much for the further constructive comments and insights. Below are our responses:
> > > > > > >
> > > > > > > ### Q1. Thank you for the clarification, as I was not aware that the embeddings you use are already normalized. I would advise you to mention this detail in the paper as this ensures the nearest neighbors are the same regardless of using L2 or cosine.
> > > > > > >
> > > > > > > **Answer:**
> > > > > > >
> > > > > > > Thank you so much for your very constructive advice! We have modified our description for this part **in lines 420-422** of the latest revision, as below：
> > > > > > >
> > > > > > > > Since our sentence embeddings are normalized, cosine similarity or Euclidean distance can be used. We follow the common practice (Douze et al., 2024) of using Euclidean distance, i.e., IndexFlatL2 in faiss, as the similarity measurement.
> > > > > > >
> > > > > > > ---
> > > > > > >
> > > > > > >
> > > > > > > ### Q2: Isn't this expected since in the RAW mode you're using the output of the final layer for fine-tuning? The intuition is that if you fine-tune any layer it's going to achieve higher scores than other layers, isn't it?
> > > > > > >
> > > > > > > **Answer:**
> > > > > > >
> > > > > > > 1) Yes. As discussed in previous answers, the last layer of LLM and MLM is crucial for capturing sentence-level semantics (as shown in Figure 3), so we set the weight of the last layer to 1.0.
> > > > > > >
> > > > > > > 2) Yes. According to our experiments, fine-tuning shallow layers can improve performance. However, existing work neglects the sentence embeddings of shallow layers, which is a research gap. To address this research gap, we propose two designs: learn-to-express and learn-to-compress. These two designs enable ESE to present superior scalability and effectiveness for sentence embedding and its downstream applications.
> > > > > > >
> > > > > > > Please kindly let us know whether we have properly addressed your comments. We shall follow up accordingly. Thank you again for your kind comments.
> > > > > > >
> > > > > > > ---
> > > > > > >
> > > > > > > ### Q3
> > > > > > > > I do not fully agree with your statement. This approach ONLY contradicts training efficiency, while inference efficiency (for the model, not the index) isn't affected. I don't think the storage requirement is a valid point, as you can easily overcome this by finding out how big of a model you can use for your use case, and only then fine-tune models up to the allowed size. Besides, you can only save the best model in each of these iterations, requiring only storage for a single model, instead of N. However, I do agree that using your method you can achieve training and inference efficiency at the same time, and you don't need to be aware of the possible use case model sizes ahead.
> > > > > > >
> > > > > > > **Answer:**
> > > > > > >
> > > > > > > We appreciate the reviewer’s comments and insights about the training efficiency and storage requirements. We agree with the reviewer that for a single, known use case, one could finetune a model of the exact required size. As reviewer also agrees with that we do not need to be aware of the possible use case model sizes ahead. We hope to emphasize the scalability and applicability that our model provides. Let’s consider a practical scenario:
> > > > > > >
> > > > > > > A company deploys LM-based services across different applications. Some applications such as real-time content searching and retrieval, require quick responses, thus need small and efficient models. More sophisticated applications like chatbots will require deeper semantic understanding and larger models. Additionally, model applications will need varying model scales due to different device capacities and resource constraints.
> > > > > > >
> > > > > > > Traditional approaches need to train and maintain separate models for each scenario. We must retrain all those models if any knowledge needs to be updated or any security issues have to be fixed. With ESE, one single trained model that dynamically scales will allow flexible deployment across different computational budgets. Whenever we need to update the models, we only need to train the model once.
> > > > > > >
> > > > > > > This becomes particularly significant when working with larger models like LLaMA. The cost and complexity of maintaining multiple differently-sized versions of models would be substantial, both in terms of storage and operational overhead.
> > > > > > > As to the statement on inference efficiency, we will revise it accordingly following your kind comment. Thank you again for your professional view!

---

> ### Author Response · Authors · 2024-11-21
> **Reply to the further comments of Reviewer 527o | Part 2**
>
> **Dear Reviewer 527o**,
>
> **Thank you very much for the further constructive comments and insights. Below are our responses:**
>
> ---
>
> ### Q4: Still, I do not think your arguments resolve the crux of the issue. Your method tunes inner layers directly, while RAW and MRL do not. I have a feeling this is actually comparing probing and fine-tuning. Can you explain why would we even expect RAW or MRL to achieve high-scoring inner representations if they are not directly tuned?
>
> **Answer:**
>
> We appreciate the reviewer's comment about the methodological difference between tuned and untuned inner representations. This actually highlights precisely the research gap our work aims to address.
>
> Existing approaches like RAW and MRL follow the conventional setting established in [1,2,3], which exclusively utilizes the last layer and does not optimize intermediate layers at all. This creates an inherent limitation in model scalability that practitioners cannot effectively leverage shallower layers even when computational resources are constrained. Our comparisons serve to highlight this limitation: **the poor performance of shallow layers in existing models is not a criticism of these methods, but rather evidence of an unaddressed need in the field**.
>
> ESE is the first work to specifically identify and address this scalability gap for sentence embeddings. By introducing mechanisms to optimize intermediate layers through a unified training process, we enable effective utilization of shallow layers when needed and flexible depth scaling without separate optimization.
>
> The baseline comparisons thus demonstrate both the existence of this research gap and ESE's effectiveness in bridging it.
>
> **Reference:**
>
> [1] Shitao Xiao, Zheng Liu, Peitian Zhang, and Niklas Muennighoff. C-pack: Packaged resources to advance general chinese embedding, 2023.
>
> [2] Xianming Li and Jing Li. Aoe: Angle-optimized embeddings for semantic textual similarity. In Pro- ceedings of the 62nd Annual Meeting of the Association for Computational Linguistics (Volume 1: Long Papers), pp. 1825–1839, 2024a.
>
> [3] Aditya Kusupati, Gantavya Bhatt, Aniket Rege, Matthew Wallingford, Aditya Sinha, Vivek Ramanujan, William Howard-Snyder, et al. Matryoshka representation learning. In Advances in Neural Information Processing Systems, volume 35, pp. 30233–30249. Curran Associates, Inc., 2022.
>
> ---
>
> ### Q5: Isn't this vector indexing efficiency already achieved by MRL? However, I am aware that your method is more efficient in the model inference phase, as MRL does not provide a mechanism for layer truncation.
>
> **Answer:**
>
> Thank you for your insightful comments. It is correct that MRL already enables flexible embedding dimensions for vector indexing. We focus more on inference efficiency. Here we did not mean to overclaim the contribution to the indexing efficiency. We will revise this part to avoid any confusion. Thank you again for your kind suggestion!
>
> ---
>
> ### Q6
> > Originally, I was most interested in how inference affects Figure 6a when I first asked this question, however this does not clarify much for me. I'm confused about the time taken for MRL for different layer indexes. If you're using MRL for the training regime, doing inference up to a certain layer, and finally slicing the embedding up to a certain dimension, shouldn't the inference times be roughly the same for MRL and ESE? The current plot for MRL looks like you're doing inference using the whole model, and only then choosing a certain layer and slicing the embedding.
>
> **Answer:**
>
> We sincerely thank the reviewer for this crucial observation about the inference time measurements in Figure 6a. Initially, we did not make a direct layer-wise timing comparison, because MRL was not designed to utilize intermediate layers for inference. When using MRL, one must use the embedding from the final layer rather than the intermediate layers. We agree with the reviewer that our original measurement for MRL’s inference time needs improvement.
> Therefore, we improved Figure 6a (**for easier reference, now shown as Figure 10 in Appendix C, will replace Figure 6a in the final version**) to show both the inference times and STS performance at different layers. The new visualization more clearly demonstrates that ESE can effectively improve shallow layer representations while maintaining comparable inference times, underscoring the scalability of our approach.
>
> ---
>
> **We sincerely thank reviewer 527o again for the thorough review and all the constructive comments on our work.**
>
> **We hope Reviewer 527o could consider raising the score accordingly if we have addressed your concerns. If you have any further concerns, please don't hesitate to let us know. We look forward to your further comments!**

---

> > ### Author Response · Authors · 2024-11-23
> > **Look forward to reviewer 527o’ reply**
> >
> > Dear reviewer 527o,
> >
> > Your earlier replies and comments are highly appreciated. We worry that you may miss the previous notifications and hope to draw your kind attention that we have responded to address your new concerns. We would greatly appreciate your evaluation of our new responses and welcome any additional feedback to enhance the quality of our work further, which does mean a lot to us!
> >
> > Thank you so much for your time and sorry for the inconvenience caused!
> >
> > Best regards,
> >
> > Authors

---

> > > ### Comment · Reviewer_527o · 2024-11-25
> > >
> > > > Yes. As discussed in previous answers, the last layer of LLM and MLM is crucial for capturing sentence-level semantics (as shown in Figure 3), so we set the weight of the last layer to 1.0.
> > >
> > > I'm trying to say that you're seeing the MLMs (and LLMs) last layer being most performant in RAW mode precisely because you are directly tuning them. If you were to directly tune a different layer, I think you would achieve the best results with that particular layer. If you then extend your design principle to the loss function, that specific layer would be left unweighted. The overall design is sound regardless, but if you want to comment on this remark, feel free to do so.
> > >
> > > > We appreciate the reviewer's comments and insights about the training efficiency and storage requirements. We agree with the reviewer that for a single, known use case, one could finetune a model of the exact required size. As reviewer also agrees with that we do not need to be aware of the possible use case model sizes ahead. We hope to emphasize the scalability and applicability that our model provides. Let's consider a practical scenario:
> > >
> > > > A company deploys LM-based services across different applications. Some applications, such as real-time content searching and retrieval, require quick responses and, thus, need small and efficient models. More sophisticated applications like chatbots will require deeper semantic understanding and larger models. Additionally, model applications will need varying model scales due to different device capacities and resource constraints.
> > >
> > > > Traditional approaches need to train and maintain separate models for each scenario. We must retrain all those models if any knowledge needs to be updated or any security issues have to be fixed. With ESE, one single trained model that dynamically scales will allow flexible deployment across different computational budgets. Whenever we need to update the models, we only need to train the model once.
> > >
> > > > This becomes particularly significant when working with larger models like LLaMA. The cost and complexity of maintaining multiple differently-sized versions of models would be substantial, both in terms of storage and operational overhead. As to the statement on inference efficiency, we will revise it accordingly following your kind comment. Thank you again for your professional view!
> > >
> > > This is a very insightful comment, and it resolves my concerns. I would appreciate it if you could add a similar comment on the engineering differences and gaps between current approaches and your approach in the final version of the paper.
> > >
> > > > We appreciate the reviewer's comment about the methodological difference between tuned and untuned inner representations. This actually highlights precisely the research gap our work aims to address.
> > >
> > > > Existing approaches like RAW and MRL follow the conventional setting established in [1,2,3], which exclusively utilizes the last layer and does not optimize intermediate layers at all. This creates an inherent limitation in model scalability that practitioners cannot effectively leverage shallower layers even when computational resources are constrained. Our comparisons serve to highlight this limitation: the poor performance of shallow layers in existing models is not a criticism of these methods, but rather evidence of an unaddressed need in the field.
> > >
> > > > ESE is the first work to specifically identify and address this scalability gap for sentence embeddings. By introducing mechanisms to optimize intermediate layers through a unified training process, we enable effective utilization of shallow layers when needed and flexible depth scaling without separate optimization.
> > >
> > > > The baseline comparisons thus demonstrate both the existence of this research gap and ESE's effectiveness in bridging it.
> > >
> > > As for the previous comment, this is a great way to explain the research gap, and it resolves all of my concerns. I suggest adding something similar in the final version of the paper.
> > >
> > > > Therefore, we improved Figure 6a (for easier reference, now shown as Figure 10 in Appendix C, will replace Figure 6a in the final version) to show both the inference times and STS performance at different layers. The new visualization more clearly demonstrates that ESE can effectively improve shallow layer representations while maintaining comparable inference times, underscoring the scalability of our approach.
> > >
> > > I appreciate the honesty in your reply. The new figure addresses my concerns and confirms the results I expected.
> > >
> > > > We hope Reviewer 527o could consider raising the score accordingly if we have addressed your concerns. If you have any further concerns, please don't hesitate to let us know. We look forward to your further comments!
> > >
> > > I think you have taken the review process very seriously and that you have addressed the issues I raised accordingly. Therefore, I have decided to raise my rating to 6: marginally above the acceptance threshold.

---

> > > > ### Author Response · Authors · 2024-11-26
> > > > **Thank you very much!**
> > > >
> > > > We sincerely thank Reviewer 527o for raising our score to 6!
> > > >
> > > > Your detailed and constructive feedback has been invaluable to our work. The thoroughness of your review deeply impressed us; we had never received such a comprehensive review before. We heartily appreciate the time and expertise you dedicated to this process. We will incorporate your comments into our final version.
> > > >
> > > > Thank you very much for your support!

---

### Official Review · Reviewer_5KSE · 2024-11-09

**Soundness:** 2
**Presentation:** 3
**Contribution:** 2
**Rating:** 5
**Confidence:** 4

**Summary:**

The paper proposes an approach for generating sentence embeddings that are scalable both in terms of model depth and embedding size. The proposed Espresso Sentence Embeddings (ESE) model incorporates two key processes: the learn-to-express process and the learn-to-compress process. The learn-to-express process aims to encode crucial latent representations in shallow layers, while the learn-to-compress process uses Principal Component Analysis (PCA) to condense essential features into the initial dimensions of the embeddings. The paper demonstrates that ESE can produce high-quality embeddings that are competitive with or surpass existing methods on standard benchmarks for semantic textual similarity (STS) and retrieval-augmented generation (RAG) tasks, particularly at reduced model depths and embedding sizes.

**Strengths:**

1. The paper is well-organized and easy to follow.
2. Experimental results are clearly presented, demonstrating the effectiveness of the proposed ESE method.

**Weaknesses:**

1. Lack of Comparison with Strong Baselines like Llama 3.1
2. The paper does not include comparisons with OpenAI's embedding models, which are considered significant benchmarks in the field of NLP.

**Questions:**

Please refer to the weakness section.

---

> ### Author Response · Authors · 2024-11-13
> **Response to Reviewer 5KSE**
>
> We would like to thank Reviewer 5KSE for your helpful comments. Below are our answers to your concerns.
>
> ## Concern 1: Lack of Comparison with Strong Baselines like Llama 3.1
>
> **Answer:**
>
> Our work focuses on scalable sentence embeddings. **Our sentence embedding model builds upon powerful transformers**, including both bi-encoder transformers (BERT, RoBERTa) and large language models (LLaMA, Qwen).
> To establish our model's competitive performance, **we have carefully selected strong and well-recognized baselines**, including:
>
> 1. bge-base-en-v1.5 (>3M monthly downloads on HuggingFace)
> 2. UAE-Large-V1 (Outperforming OpenAI embeddings on MTEB, >1M monthly downloads on HuggingFace)
> 3. Popular LLM-based backbones: Qwen and LLaMA2
>
> Using these powerful backbones, we conducted comprehensive comparisons with RAW and MRL to demonstrate the scalability and effectiveness of our proposed model.
>
> Regarding LLaMA 3.1: We acknowledge its recent release and **appreciate the suggestion to include it in our comparisons**. While the rapid development of LLMs makes it challenging to update to the latest backbones continuously, **we are currently working on extending our experiments to include LLaMA 3.1. It will take some time. We expect to complete these additional experiments and update our results within next few days.** **Please stay tuned!**
>
> ---
>
> ## Concern 2: The paper does not include comparisons with OpenAI's embedding models, which are considered significant benchmarks in the field of NLP.
>
> **Answer:**
>
> The reasons why we did not compare with OpenAI's embeddings are as follows:
>
> First, OpenAI's embedding models are closed-source, making it impossible to apply our proposed ESE training methodology to them for a direct comparison.
>
> Second, OpenAI's API only provides access to the final layer embeddings, preventing any analysis or comparison of shallow layer representations. This limitation is particularly relevant since improving shallow layer performance is a key contribution of the proposed ESE approach.
>
> Third, our chosen backbone **UAE-Large-V1 is a strong open-source model that outperforms OpenAI's text-embedding on the MTEB benchmark [1].** Following your suggestion, we extend our Table 1 to add the last layer performance of OpenAI text-embedding-3, as follows:
>
> | Model                         | STS12 | STS13 | STS14 | STS15 | STS16 | STSB  | Sick-R | Average |
> |-------------------------------|:-----:|-------|-------|-------|-------|-------|--------|---------|
> | OpenAI text-embedding-3-large | 71.98 | 85.82 | 80.5  | 87.51 | 84.48 | 82.34 | 79.18  | 81.69   |
> | UAE-Large-V1                  | 79.09 | 89.62 | 85.02 | 89.51 | 86.61 | 89.06 | 82.10  | 85.86   |
> | ESE-Large                     | 79.64 | 90.40 | 85.76 | 90.33 | 86.64 | 88.54 | 81.09  | **86.06**   |
>
>
> The results demonstrate that ESE can outperform strong baselines. Due to the aforementioned API limitations regarding shallow layer access, a comprehensive comparison of shallow layer emebddings with OpenAI's model remains unfeasible.
>
> ---
>
>
> Thank you again for your very kind effort and professional input! We will include the comparison between our results and OpenAI’s embedding in our final version.
>
> **We hope Reviewer 5KSE could consider raising the score accordingly if we have addressed your concerns. If you have any further concerns, please don't hesitate to let us know.**
>
>
> ---
>
>
> **Reference:**
>
> - [1] Muennighoff, N., Tazi, N., Magne, L., & Reimers, N. (2022). MTEB: Massive text embedding benchmark. arXiv preprint arXiv:2210.07316.

---

> ### Author Response · Authors · 2024-11-16
> **Response to Reviewer 5KSE [LLaMA3.1 Results]**
>
> Following your advice, we compare our approach with LLaMA3.1 8B. The results are presented **in Appendix Section A (LLAMA3.1 STS RESULTS) of the latest revision of PDF**. The LLaMA3.1 results are similar to the LLaMA2 results. The difference is that LLaMA3.1 can marginally improve performance. **The experimental results show that ESE significantly improves the performance of shallow layers while consistently outperforming baselines in the final layer.**
>
> We will include the comparison between our results and LLaMA3.1 in our final version.
>
> **We hope Reviewer 5KSE could consider raising the score accordingly if we have addressed your concerns. If you have any further concerns, please don't hesitate to let us know.**

---

> > ### Author Response · Authors · 2024-11-27
> > **Look forward to your reply!**
> >
> > Dear Reviewer **5KSE**,
> >
> >
> > We would like to draw your kind attention to our **completed additional experiments comparing ESE with LLaMA 3.1 and OpenAI models**. **We await your valuable feedback on whether these new results address your concerns.** Your further engagement will be greatly appreciated.
> >
> > Additionally, we would like to ensure clarity about ESE's core contributions:
> >
> > **Research Gap:**
> >
> > Current approaches to sentence embeddings face two key limitations, i.e., underutilized shallow layers, reducing deployment flexibility, and limited scalability requiring multiple optimized models.
> >
> > **Our Contribution:**
> >
> > 1. We propose a novel Information Compression mechanism ESE (Espresso Sentence Embeddings):
> >    - Introduces information compression to concentrate features in initial embeddings, eliminating the need for multiple embedding size optimizations
> >    - Implements weighted shallow layer optimization to enable vertical scalability
> >
> > 2. Efficient Two-Process Framework:
> >     - Learn-to-Express: Optimizes each layer's embeddings for diverse applications (e.g., information retrieval, semantic textual similarity)
> >     - Learn-to-Compress: Aligns top-k representations with compressed-k versions to concentrate essential features in initial segments, enabling efficient dynamic embedding sizes
> >
> > 3. Practical Impact:
> >    - Single trainable model supporting multiple deployment configurations
> >    - Dynamic adjustment to computing constraints
> >    - Improved efficiency in real-world applications
> >
> > The additional experimental results further validate these capabilities across different model scales and architectures. We look forward to any feedback on these new findings or our core contributions.
> >
> > Best,
> >
> > Authors

---

> > > ### Author Response · Authors · 2024-12-01
> > > **Response to Reviewer 5KSE**
> > >
> > > Dear Reviewer 5KSE,
> > >
> > > Thank you again for your valuable comments on our work. **As the rebuttal deadline approaches, we would greatly appreciate hearing from you regarding whether our responses have adequately addressed their concerns.**
> > >
> > > As requested by the reviewer, we have provided additional experimental results by comparing ESE with the state-of-the-art LLaMA3.1 (reported in Appendix Section A) and OpenAI embedding. For your easier reference, our detailed responses can be found here:
> > >
> > >
> > > | Topic | Response Link |
> > > |-------|---------------|
> > > | LLaMA3.1 Comparison | [Response](https://openreview.net/forum?id=plgLA2YBLH&noteId=89U0K3XzpH) |
> > > | OpenAI Comparison | [Response](https://openreview.net/forum?id=plgLA2YBLH&noteId=nKbNepAOke) |
> > >
> > >
> > > Your continued engagement in this discussion is crucial for further improving our work. We welcome any additional feedback or suggestions you may have. Thank you so much!
> > >
> > >
> > > Best,
> > >
> > > Authors.

---

### Author Response · Authors · 2024-11-21
**Reponse to All Reviewers**

Dear All Reviewers,

Thank you for your thorough review and valuable feedback on our work.

We have carefully addressed all comments and concerns raised during the review process. We would greatly appreciate your evaluation of our responses and welcome any additional feedback to enhance the quality of our work further.

Thank you for your time and consideration.

Best regards,

Authors.

---

### Author Response · Authors · 2024-11-25
**Look forward to your reply!**

Dear Reviewers **5KSE**, **876W**, and **527o**:

Thank you for your valuable comments on our work. As the rebuttal deadline approaches, **we would greatly appreciate hearing from reviewers 5KSE and 876W regarding whether our responses have adequately addressed their concerns.**

We are particularly grateful to reviewer 527o for the comprehensive and professional review and detailed suggestions that have significantly improved our work. **We look forward to receiving feedback from 527o on our responses to the follow-up questions.**

Below is a summary table of our revisions addressing the concerns raised by reviewers 5KSE, 876W, and 527o:

| **Update**                                                                    |         **Section**        | **Reviwers** |
|---------------------------------------------------------------------------|:----------------------:|----------|
| Compare with LLaMA3.1                                                     | Appendix Section A     | 5KSE     |
| Compare with OpenAI                                                       | Openreview Comment Box | 5KSE     |
| Explanation for model design and real-world applications                      | Openreview Comment Box | 876W     |
| Addressing the followed-up concerns related to efficiency and scalability | Openreview Comment Box | 527o     |



- **5KSE:** [llama3.1](https://openreview.net/forum?id=plgLA2YBLH&noteId=89U0K3XzpH) and
[openai](https://openreview.net/forum?id=plgLA2YBLH&noteId=nKbNepAOke)

- **876W:** [part 1](https://openreview.net/forum?id=plgLA2YBLH&noteId=rTRAsIZ2cH)
and [part 2](https://openreview.net/forum?id=plgLA2YBLH&noteId=x7Q5rzEYUn)

- **527o follow-up replies:** [part 1](https://openreview.net/forum?id=plgLA2YBLH&noteId=VVOSWg9Bvj)
and [part 2](https://openreview.net/forum?id=plgLA2YBLH&noteId=DzYR7V90uP)

---

Your continued engagement in this discussion is crucial for further improving our work. We welcome any additional feedback or suggestions you may have. Thank you so much!

---

### Meta-Review · Area_Chair_Rojy · 2024-12-23

**Metareview:**

The paper proposes Espresso Sentence Embeddings (ESE) for scalable sentence embedding learning, aiming to address the challenges of fixed embedding sizes and model depths in existing sentence embedding methods.
Reviewers generally recognized the value of the paper's core contribution, which is enabling scalable sentence embeddings. Multiple reviewers gave positive comments on the presentation and experiments, as well as the clear research objectives and well-designed training methodology.  Although this paper is on the borderline, most of the concerns are fixed in the discussion period. Two negative reviewers do not give any response after rebuttal, and after reading their review, I regard them as invalid.

**Additional Comments On Reviewer Discussion:**

There were concerns regarding the comparison with strong baselines (5KSE), the focus and clarity of the contribution statement, completeness of the related work, and some design choices (527o), the complexity of the design and real-world applications (876W), and the generalizability of the experimental scope (XogJ).   Most of the concerns are fixed in the discussion period.

---

### Decision · Program_Chairs · 2025-01-22

Accept (Poster)